# Abstracting Robot Manipulation Skills via Mixture-of-Experts Diffusion Policies

**Ce Hao**[*†1,4], **Xuanran Zhai**[*1], **Yaohua Liu**[3], **Harold Soh**[†1,2]

[1] School of Computing, National University of Singapore; [2] Smart Systems Institute, NUS;
[3] Guangdong Institute of Intelligence Science and Technology; [4] Beijing Zhongguancun Academy
[*] Equal contribution. [†] Emails: `cehao@u.nus.edu` and `harold@nus.edu.sg`

## Abstract

Diffusion-based policies have recently shown strong results in robot manipulation, but their extension to multi-task scenarios is hindered by the high cost of scaling model size and demonstrations. We introduce **Skill Mixture-of-Experts Policy** (**SMP**), a diffusion-based mixture-of-experts policy that learns a compact orthogonal skill basis and uses sticky routing to compose actions from a small, task-relevant subset of experts at each step. A variational training objective supports this design, and adaptive expert activation at inference yields fast sampling without oversized backbones. We validate SMP in simulation and on a real dual-arm platform with multi-task learning and transfer learning tasks, where SMP achieves higher success rates and markedly lower inference cost than large diffusion baselines. These results indicate a practical path toward scalable, transferable multi-task manipulation: learn reusable skills once, activate only what is needed, and adapt quickly when tasks change.

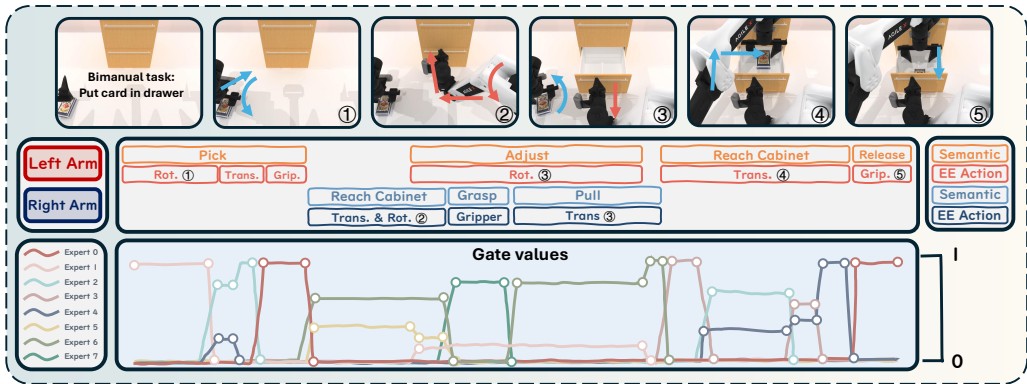

Figure 1: **Overview of SMP.** *Top:* Bimanual rollout of "put card in drawer" with key steps (1)–(5). *Middle:* Skill decomposition by arm and phase: the state-adaptive orthonormal skill basis and sticky routing yields spatial specialization (left/right), and organizes behavior into pick, adjust, reach, and release with corresponding end-effector (EE) actions. *Bottom:* Gate values over time show sparse, phase-consistent activation—only a few experts are active per step for efficient sampling.

## 1 Introduction

Recent advances in diffusion-based policies have demonstrated remarkable success in solving complex robot manipulation tasks (Chi et al., 2023; Hao et al., 2025). By modeling action generation as a denoising process, these approaches achieve high success rates and robustness in single-task settings. However, how to effectively generalize such policies to diverse multi-task scenarios remains a crucial open challenge in robotics. A common strategy pursued in recent research is to scale up the size of policy networks (Liu et al., 2024), motivated by the scaling laws that enable large models to interpolate across unseen tasks. While promising, this paradigm comes at a steep cost: the inference speed of oversized models is often impractical for real-time manipulation (Mei et al., 2024) and the required demonstration datasets may grow exponentially with task diversity. Thus, enabling effec-

tive multi-task generalization under moderate model size and sampling latency is vital for real-world robotic applications.

An alternative line of work seeks generalization through skill-based policy learning (Liang et al., 2024; Wu et al., 2025), which abstracts task-invariant skills from demonstrations and reuses them across tasks. Prior approaches, such as information-theoretic diversity regularization (*e.g.*, DIAYN (Eysenbach et al., 2018)) and goal-conditioned hierarchical RL (Liu et al., 2022), are designed primarily for exploration in sparse-reward environments rather than efficient skill abstraction for manipulation. More recently, mixture-of-experts (MoE) architectures have been applied to robot diffusion policies (Wang et al., 2024), where a large feedforward backbone is replaced by smaller expert modules. However, existing MoE formulations do not explicitly disentangle and represent reusable manipulation skills, limiting their interpretability and transferability (Yang et al., 2025).

In this paper, we introduce **Skill Mixture-of-Experts Policy** (**SMP**) (Fig. 1), a diffusion-based mixture-of-experts policy that performs skill abstraction in a *locally whitened* action space. Here, a *skill* denotes a state-adaptive orthogonal action primitive—locally disentangled in the current action geometry—whose consistent activation patterns across time and tasks form higher-level roles. Rather than blending unconstrained experts, SMP learns these state-adaptive skills and employs slowly varying (*sticky*) gates, yielding disentangled, phase-consistent behaviors. We develop a principled variational objective that combines reconstruction in the whitened basis and gate regularization, and distill a *state-only* router. At inference, an *adaptive expert activation* mechanism selects a compact subset of experts. Together, these choices produce compact, reusable, and transferable skills that improve multi-task generalization and sampling efficiency, enabling real-time bimanual manipulation.

We validate SMP in both simulation (Chen et al., 2025; Grotz et al., 2024) and real-world bimanual manipulation tasks (Fu et al., 2024). Across multi-task evaluations, SMP achieves consistently good performance with lower inference cost than strong diffusion baselines. Analyses show that the orthonormal skill basis and sticky routing yield stable behavior with fewer gate switches and cross-task skill reuse. Adaptive expert activation maintains policy quality while substantially reducing active parameters and latency. In transfer learning, reusing the compact skill set enables effective few-shot adaptation to new tasks.

In summary, the contributions of this paper are as follows:

- We propose SMP, a diffusion-based mixture-of-experts framework that explicitly abstracts reusable manipulation skills via a state-dependent orthonormal action basis with sticky routing, improving performance across multiple tasks.
- We design an adaptive expert activation strategy that dynamically selects a compact subset of experts at inference time, reducing computational cost while maintaining action sampling accuracy.
- We validate SMP in bimanual manipulation tasks—including multi-task and transfer settings—demonstrating high success rates and lower inference cost than strong diffusion baselines.

## 2 BACKGROUND AND RELATED WORK

**Robot Manipulation Policies**. Diffusion-based generative models have recently become a strong paradigm for robot manipulation, where policies such as Diffusion Policy (Chi et al., 2023) and flow-matching variants (Zhang et al., 2025) generate actions via iterative denoising or continuous-time dynamics. These methods achieve strong single-task performance and stable training, and have been extended to longer horizons using trajectory or decision transformers (Chen et al., 2021). Vision–language–action (VLA) models (Zitkovich et al., 2023; Kim et al., 2024) further improve generalization by leveraging large-scale multimodal pretraining, while other works combine keypoint-conditioned representations (Sundaresan et al., 2023) or hybrid controllers such as MPC (Zhao et al., 2024). Although effective, these methods generally rely on scaling up data and model capacity to gain transferability, which introduces prohibitive computational costs for real-time control.

**Generalizable Skill Abstraction**. A broad line of work aims to reuse task-agnostic motion primitives across tasks. Unsupervised skill discovery methods maximize mutual information or related diversity criteria to induce distinct behaviors without rewards (e.g., DIAYN (Eysenbach et al., 2018)), while hierarchical/option frameworks learn temporally extended actions to simplify long-horizon control (Liu et al., 2022). A complementary thread learns latent skill spaces from demonstrations via variational or discrete codebook models, enabling a policy to condition on compact skill iden-

tifiers during execution (Liang et al., 2024; Wu et al., 2025). More recently, mixture-of-experts architectures (Reuss et al., 2024) have been adopted to scale capacity and route computation across modules within diffusion-based policies (Huang et al., 2024; Wang et al., 2024). Despite progress, most approaches do not explicitly decorrelate the action space or enforce phase stability—skills often remain entangled, routing can switch rapidly, and reuse across bimanual roles is opaque.

## 3 PRELIMINARIES

**Problem Formulation**. We define a set of robot manipulation tasks $\mathcal{M} = \{M_\kappa\}_{\kappa=1}^m$ with $m$ sub-tasks $M_\kappa$. All tasks share the same robot state space $s \in \mathcal{S}$ and action space $a \in \mathcal{A}$. Note that the state space includes both observations and the task index $\kappa$, which serves as a task identifier to distinguish and indicate which sub-task the policy should execute. We collect trajectory demonstrations in each task $\mathcal{D}_\kappa = \{\mathcal{T}_{\kappa,j}\}$ and each trajectory contains a series of consecutive state and action pairs $\mathcal{T}_{\kappa,j} = \{(s_i, a_i)\}$. In the multi-task robot manipulation problem, a manipulation policy $p(a|s)$ is trained in all tasks $\mathcal{M}$ to imitate the demonstrations $\mathcal{D}$. We evaluate the success rate of completing each task $M_\kappa$ and computation cost (i.e. model size, computation time) in the policy training and sampling processes.

**Diffusion Policy.** We adopt diffusion-based generative modeling for action generation, where a "clean" action $a_0$ is viewed as the endpoint of a latent Markov chain $a_{1:\mathfrak{T}}$ of the same dimensionality. The model defines $p_\theta(a_0) = \int p_\theta(a_{0:\mathfrak{T}}) \, da_{1:\mathfrak{T}}$. The forward (noising) process gradually perturbs data $a_0 \sim q(a_0)$ with a variance schedule $\{\beta_\tau\}_{\tau=1}^{\mathfrak{T}}$, factorized as $q(a_{1:\mathfrak{T}} \mid a_0) = \prod_{\tau=1}^{\mathfrak{T}} q(a_\tau \mid a_{\tau-1})$ with $q(a_\tau \mid a_{\tau-1}) = \mathcal{N}(a_\tau; \sqrt{1-\beta_\tau}\, a_{\tau-1}, \beta_\tau I)$. Conditioned on the current state $s$, the reverse (denoising) process is parameterized by a Gaussian chain $p_\theta(a_{0:\mathfrak{T}} \mid s) = p(a_\mathfrak{T}) \prod_{\tau=1}^{\mathfrak{T}} p_\theta(a_{\tau-1} \mid a_\tau, s)$ with standard normal prior $p(a_\mathfrak{T}) = \mathcal{N}(0, I)$ and $p_\theta(a_{\tau-1} \mid a_\tau, s) = \mathcal{N}(\mu_\theta(a_\tau, \tau, s), \Sigma_\theta(a_\tau, \tau, s))$. Training maximizes the conditional data likelihood by minimizing a diffusion ELBO: $\mathbb{E}[-\log p(a \mid s)] \leq \mathbb{E}_q\left[-\log \frac{p_\theta(a_{0:\mathfrak{T}}|s)}{q(a_{1:\mathfrak{T}}|a_0)}\right] := \mathcal{L}_{\text{Diff}}(a, a_0)$, where $\mathcal{L}_{\text{Diff}}$ is implemented via standard DDPM (Ho et al., 2020) parameterizations of $\mu_\theta$ and $\Sigma_\theta$ (or equivalent noise-prediction losses) and supports efficient conditional sampling at test time.

## 4 METHOD: SKILL MIXTURE-OF-EXPERT POLICY

Direct mixtures of unconstrained expert outputs can overlap and become non-identifiable; many different combinations can reproduce the same action, leading to unstable routing and training. To address this, we introduce a method for skill disentanglement based on constructing a *state-adaptive orthogonal frame*. At each state, every skill is mapped to a distinct, non-overlapping direction in the action space. This ensures non-overlapping contributions, yields a well-conditioned demixing for supervising coefficients, and—combined with sticky gates—promotes sparse, stable activation of only a few skills per state. In the following, we describe the core ideas underlying our method; an overview of our framework and training pipeline is shown in Fig. 2. Additional implementation details are provided in Appendix A.

**Generative model**. At time $t$, $s_t \in \mathbb{R}^{d_s}$ denotes the state and $a_t \in \mathbb{R}^d$ the action. Let $K \ll d$ be the number of reusable skills. We write $g_t \in \Delta^{K-1}$ for the gate (simplex weights over skills, App. A.1) and $z_t \in \mathbb{R}^K$ for the per-skill coefficients. Actions are decoded through an orthonormal skill basis $B = [b_1, \ldots, b_K] \in \mathbb{R}^{d \times K}$,

$$a_t = B\left(g_t \odot z_t\right) \tag{1}$$

where $B^\top B = I_K$ and $\odot$ is elementwise product. In this orthogonal decomposition, the $i$-th skill contributes the rank-1 vector $b_i \left(g_{t,i} z_{t,i}\right)$ in $\text{span}\{b_i\}$, which ensures additive effects of each skill. While a global skill basis $B$ is appealing conceptually, in practice robot action geometry may depend on state (e.g., arm pose, contact). A fixed basis may fail to capture such variability. We therefore parameterize a state-dependent basis $B(s)$, which allows the skill frame to adjust with context while preserving orthogonality and local disentanglement.

**QR factorization for state-adaptive skill basis**. To construct the basis $B(s) \in \mathbb{R}^{d \times K}$, we first generate an unconstrained matrix $W(s) \in \mathbb{R}^{d \times K}$ with a lightweight neural network, and project it onto the Stiefel manifold using a differentiable thin–QR retraction with sign stabilization (App. A.1).

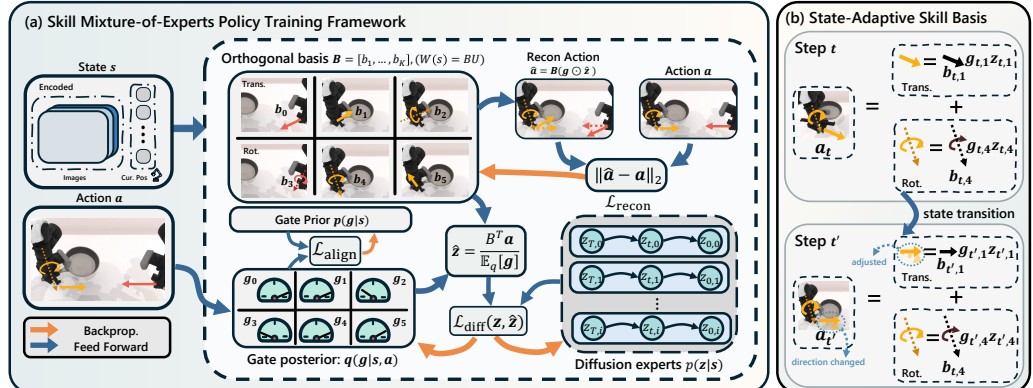

Figure 2: **Skill Mixture-of-Experts Policy (SMP) Training Framework. Left (a)**: During training, raw observations are encoded into state features, which generate an unconstrained matrix $W(s)$. A QR retraction produces a state-adaptive orthogonal basis $B(s)$. Actions are reconstructed via $B(s)(g \odot z)$, where $g$ are sticky-gated weights and $z$ are diffusion-based coefficients. The model is trained with reconstruction, diffusion, gate regularization, and alignment losses. **Right (b)**: Illustration of the state-adaptive basis across timesteps: as the robot moves, the basis vectors adjust with the state, while sticky gates preserve consistent expert roles (e.g., translation and rotation).

Concretely, for each state $s$, we compute

$$W(s) = \tilde{B} U, \quad D = \text{diag}\big(\text{sign}(\text{diag}(U))\big), \quad B(s) = \tilde{B} D, \quad \tilde{B}^\top \tilde{B} = I_K, \ U \text{ upper triangular.} \tag{2}$$

Here, $\tilde{B}$ is the orthonormal factor from the QR decomposition, and $D$ is a diagonal sign matrix that removes the column–sign ambiguity. This encourages $B(s)$ to evolve continuously with $s$ by avoiding discontinuous flips in orientation. During training, we treat $B(s)$ as the forward map, update $W(s)$ via standard optimizers, and backpropagate through the QR retraction using automatic differentiation. This formulation yields a moving orthogonal frame that adapts with task geometry, while the gating mechanism provides phase consistency. To avoid notational clutter, we omit the explicit dependence on $s$ in $B(s)$ and $W(s)$ when the context is clear.

**Sticky gates and global usage.** Manipulation typically progresses through quasi-stationary phases, so the skill gate $g_t \in \Delta^{K-1}$ should change slowly rather than chatter at every step, while still avoiding collapse to a small subset of skills. We formalize this intuition via "sticky" Dirichlet Markov dynamics (App. A.1):

$$\vartheta \sim \text{Dir}(\alpha \, \mathbf{1}), \quad g_1 \sim \text{Dir}(\alpha_0 \, \vartheta), \quad g_t \sim \text{Dir}\big(\kappa \, g_{t-1} + \alpha_0 \, \vartheta\big), \ t \geq 2. \tag{3}$$

Here $\vartheta \in \Delta^{K-1}$ is a global usage vector capturing overall skill prevalence; the initial gate $g_1$ is drawn around $\vartheta$; and subsequent gates $g_t$ blend persistence from the previous gate with a mild pull toward global usage. The hyperparameters have intuitive roles: $\kappa > 0$ controls temporal stickiness (larger values yield longer, phase-like segments), $\alpha_0 > 0$ anchors the process to $\vartheta$ to avoid degeneracy, and $\alpha > 0$ sets the diffuseness of the global prior (larger values encourage more balanced usage). This model yields piecewise-constant skill activations while maintaining broad yet non-uniform utilization across tasks.

## 4.1 Variational Lower-Bound and Loss Formulation

**Variational inference for joint generation.** We formalize SMP training via variational inference. We group the latents into gates and global usage $(\vartheta, g_{1:T})$ and skill coefficients $z_{1:T}$. The trajectory-level model factorizes as:

$$p_\theta(\vartheta, g_{1:T}, z_{1:T}, a_{1:T} \mid s_{1:T}) = \underbrace{p\big(a_{1:T} \mid g_{1:T}, z_{1:T}, s_{1:T}, B\big)}_{\text{reconstruction: } a_t = B \, (g_t \odot z_t)} \underbrace{p(\vartheta, g_{1:T} \mid s_{1:T})}_{\text{gates}} \underbrace{p(z_{1:T} \mid s_{1:T})}_{\text{coefficients}}. \tag{4}$$

Here the first term enforces reconstruction in the local whitened basis; the second term is the sticky Dirichlet prior with a global usage vector, $p(\vartheta, g_{1:T} \mid s_{1:T}) = p(\vartheta) \, p(g_1 \mid \vartheta) \prod_{t=2}^{T} p(g_t \mid g_{t-1}, \vartheta)$; and the third term adopts diffusion priors for coefficients, $p(z_{1:T} \mid s_{1:T}) = \prod_{t=1}^{T} p(z_t \mid s_t)$. For

inference, we adopt an amortized, factorized posterior,

$$q(\vartheta, g_{1:T}, z_{1:T} \mid a_{1:T}, s_{1:T}) = \underbrace{q(\vartheta, g_{1:T} \mid a_{1:T}, s_{1:T})}_{\text{Dirichlet amortizers}} \underbrace{q(z_{1:T} \mid a_{1:T}, s_{1:T})}_{\text{coefficient posteriors}}. \quad (5)$$

Applying Jensen's inequality to Eqn. (4) and (5) yields:

$$\log p_\theta(a_{1:T} \mid s_{1:T}) = \log \int q(\cdot) \frac{p_\theta(\vartheta, g_{1:T}, z_{1:T}, a_{1:T} \mid s_{1:T})}{q(\vartheta, g_{1:T}, z_{1:T} \mid a_{1:T}, s_{1:T})} \, d\vartheta \, dg \, dz \quad (6)$$

$$\geq \underbrace{\mathbb{E}_q\Big[\log p\big(a_{1:T} \mid g_{1:T}, z_{1:T}, s_{1:T}, B\big)\Big]}_{\text{reconstruction term: } \mathcal{L}_{\text{recon}}} - \underbrace{D_{\text{KL}}\Big(q(\vartheta, g_{1:T} \mid a_{1:T}, s_{1:T}) \, \Big\| \, p(\vartheta, g_{1:T} \mid s_{1:T})\Big)}_{\text{gate/global-usage regularization: } \mathcal{L}_{\text{gate}}}$$

$$- \underbrace{D_{\text{KL}}\Big(q(z_{1:T} \mid a_{1:T}, s_{1:T}) \, \Big\| \, p(z_{1:T} \mid s_{1:T})\Big)}_{\text{coefficient regularization: } \mathcal{L}_{\text{coeff}}}.$$

In summary, Eqn. (6) separates training into three components: (i) a reconstruction term realized by synthesis in the local basis (implemented via the coefficient-space diffusion, described below), (ii) a gate/global-usage regularizer $\mathcal{L}_{\text{gate}}$ that pulls the amortized gate posterior toward the sticky-gate prior, and (iii) a coefficient regularizer $\mathcal{L}_{\text{coeff}}$ that aligns coefficient posteriors with their diffusion priors.

**Reconstruction via coefficient-space diffusion.** We decode actions $\hat{a}_t = B\big(g_t \odot z_{0,t}\big)$ and place a small-variance Gaussian around it, $p(a_t \mid z_{0,t}, g_t, s_t, B) = \mathcal{N}(\hat{a}_t, \sigma_a^2 I)$, yielding $\mathcal{L}_{\text{recon}} = \frac{1}{2\sigma_a^2} \sum_{t=1}^T \|a_t - \hat{a}_t\|_2^2$. To supervise coefficients while disentangling which gradients reach $B$, we form two targets and the reconstruction:

$$\underbrace{\hat{z}_{0,t}^{\text{sg}} = \frac{\bar{B}^\top a_t}{\mathbb{E}_q[g_t] + \epsilon}}_{\text{stop–gradient for diffusion}}, \, \underbrace{\hat{z}_{0,t}^{\text{rec}} = \frac{B^\top a_t}{\mathbb{E}_q[g_t] + \epsilon}}_{\text{gradient flows to } B}, \, \hat{a}_t^{\text{rec}} = B\big(g_t \odot \hat{z}_{0,t}^{\text{rec}}\big), \, \mathcal{L}_{\text{recon}} = \frac{1}{2\sigma_a^2} \sum_{t=1}^T \|a_t - \hat{a}_t^{\text{rec}}\|_2^2. \quad (7)$$

Here $\bar{B} = \text{sg}[B]$ is a stop–gradient copy and the division is elementwise ($\epsilon \approx 10^{-3}$). We use $\hat{z}_{0,t}^{\text{sg}}$ for the diffusion surrogate, $\mathcal{L}_{\text{coeff}} = \mathcal{L}_{\text{diff}}(z; \hat{z}_{0,1:T}^{\text{sg}})$, so no gradient from $\mathcal{L}_{\text{coeff}}$ updates $B$. In contrast, reconstruction uses the gradient-carrying target $\hat{z}_{0,t}^{\text{rec}}$, which propagates gradients into $B$ (and $g_t$) through both the projection $B^\top a_t$ and the decoder $B(\cdot)$. Together, $\mathcal{L}_{\text{recon}} + \mathcal{L}_{\text{coeff}}$ encourages action consistency while providing stable per-expert supervision in coefficient space, with only the reconstruction term updating the skill basis.

**Gate regularization.** We use Dirichlet distributions $q(\vartheta) = \text{Dir}(\hat{\alpha})$ and $q(g_t \mid s_t, a_t) = \text{Dir}(\hat{\beta}_t(s_t, a_t))$, and train coefficients with the DDPM loss. We also introduce a *state-only router* for deployment, $p_\phi(g_t \mid s_t) = \text{Dir}(\tilde{\beta}_\phi(s_t))$, whose distribution is aligned to the training-time gate posterior. Plugging the sticky-gate prior (Eqn. 3) into the ELBO yields:

$$\mathcal{L}_{\text{gate}} = \underbrace{D_{KL}\big(q(\vartheta) \, \| \, \text{Dir}(\alpha \mathbf{1})\big)}_{\text{global usage}} + \underbrace{D_{KL}\big(q(g_1) \, \| \, \text{Dir}(\alpha_0 \, \mathbb{E}_q[\vartheta])\big)}_{\text{initial gate}} + \quad (8)$$

$$\sum_{t=2}^T \underbrace{D_{KL}\Big(q(g_t) \, \Big\| \, \text{Dir}\big(\kappa \, \mathbb{E}_q[g_{t-1}] + \alpha_0 \, \mathbb{E}_q[\vartheta]\big)\Big)}_{\text{sticky gates}}, \, \mathcal{L}_{\text{align}} = \sum_{t=1}^T \underbrace{D_{KL}\big(q(g_t \mid s_t, a_t) \, \| \, \text{Dir}(\tilde{\beta}_\phi(s_t))\big)}_{\text{router alignment}}.$$

The three terms in $\mathcal{L}_{\text{gate}}$ impose a global usage prior, anchor the initial gate, and encourage temporal stickiness via persistence from $g_{t-1}$. The auxiliary loss $\mathcal{L}_{\text{align}}$ aligns the state-only router to the training-time gate posterior so deployment-time routing is consistent with inference.

**Mixture-of-Experts equivalence.** Our decoder is exactly an additive MoE in the local skill basis: $a_t = \sum_{i=1}^K g_{t,i} \, b_i \, z_{t,i} = B\big(g_t \odot z_t\big)$. Interpreting the $i$-th expert as the rank-1 map $f_i(s_t; z_{t,i}) = b_i z_{t,i}$ with coefficient distribution $p(z_{t,i} \mid s_t)$ yields the pushforward expert conditional $b_i(a_t \mid s_t) = \int \delta(a_t - b_i z) \, p(z \mid s_t) \, dz$, and the gate is $w_i(s_t) = g_{t,i}$. Because $\{b_i\}$ are orthonormal, each expert acts on the one-dimensional subspace $\text{span}\{b_i\}$, giving orthogonal contributions and decoupled gradients; top-$k$ routing makes the mixture sparse.

## 4.2 ADAPTIVE EXPERT ACTIVATION

**Activation with top-$k$ or coverage selection.** Evaluating all experts at every state is costly and unnecessary since only a few skill directions are typically important. At deployment we use the *state-only* router $p_\phi(g_t \mid s_t) = \mathrm{Dir}(\tilde{\beta}_\phi(s_t))$ and its mean $\bar{g}_t = \mathbb{E}[g_t \mid s_t]$ to estimate the importance of each expert. In the orthonormal basis $B = [b_1, \ldots, b_K]$ with $B^\top B = I$, we define the *mass* of expert $i$ as $m_i = \bar{g}_{t,i}^2$.

To decide which experts to activate, we score any candidate active set $S \subseteq \{1, \ldots, K\}$ by $F(S) = \sum_{i \in S} m_i$. Because $F$ is additive, the optimal set can be obtained by simply sorting experts by $m_i$ and selecting either (i) the top-$k$ experts, or (ii) the smallest prefix of experts such that the selected set captures at least a fraction $\tau_m$ of the total mass, i.e., $\frac{\sum_{i \in S} m_i}{\sum_{j=1}^{K} m_j} \geq \tau_m$, with $\tau_m \in [0.9, 0.95]$.

After choosing the active set $S_t$, we denoise only the corresponding coefficients $z_{t,S_t}$ (all others set to zero) and decode the action as $a_t = B(\bar{g}_t \odot z_t)$. This simple ranking rule is equivalent to a greedy maximization of the additive objective $F(S) = \sum_{i \in S} m_i$. It yields sparse, state-dependent activation, reducing inference cost while preserving accuracy.

## 4.3 ALGORITHM SUMMARY

We summarize SMP in Algorithms 1 and 2. Training alternates over trajectories by (i) orthonormalizing the skill basis $B = \mathrm{qrf}(W)$, (ii) inferring gates with the amortized posterior, (iii) forming two coefficient targets in the whitened basis—$\hat{z}_0^{\mathrm{sg}}$ (stop–gradient) for the diffusion surrogate and $\hat{z}_0^{\mathrm{rec}}$ (gradient-carrying) for reconstruction—and (iv) optimizing a compact objective that combines coefficient-space diffusion, a light reconstruction term, sticky-gate regularization, and router alignment. At inference, given $s_t$ we query the state-only router, select a small active set of experts via top-$k$ or greedy coverage, denoise only the selected coefficients, and decode the action via $a_t = B(\bar{g}_t \odot z_t)$, yielding sparse, low-latency control.

---

**Algorithm 1** SMP Training

1: **Input**: dataset $\mathcal{D}$; basis params $W$ ($B = \mathrm{qrf}(W)$); gate posterior $q(g \mid s, a)$; state-only router $p_\phi(g \mid s)$; diffusion experts for $z$.
2: **repeat**
3:     Sample a trajectory $(s_{1:T}, a_{1:T}) \sim \mathcal{D}$
4:     Orthonormalize $B \leftarrow \mathrm{qrf}(W)$ (sign-stabilized)
5:     Compute amortized gates $q(g_t \mid s_t, a_t)$ for $t = 1{:}T$
6:     Build two coefficient targets $\hat{z}_{0,t}^{\mathrm{sg}}, \hat{z}_{0,t}^{\mathrm{rec}}$
7:     Compute diffusion loss $\mathcal{L}_{\mathrm{coeff}} \leftarrow \mathcal{L}_{\mathrm{diff}}(z; \hat{z}_{0,1:T}^{\mathrm{sg}})$
8:     Reconstruct $\hat{a}_t^{\mathrm{rec}} = B(g_t \odot \hat{z}_{0,t}^{\mathrm{rec}})$ and $\mathcal{L}_{\mathrm{recon}}$
9:     Compute gate regularizers $\mathcal{L}_{\mathrm{gate}}$ and router alignment $\mathcal{L}_{\mathrm{align}}$
10:     Total loss: $\mathcal{L}_{\mathrm{SkillMoE}} \leftarrow \mathcal{L}_{\mathrm{coeff}} + \mathcal{L}_{\mathrm{recon}} + \mathcal{L}_{\mathrm{gate}} + \mathcal{L}_{\mathrm{align}}$
11:     Update $W$, diffusion experts, amortizers, and router via gradient descent
12: **until** converged

---

**Algorithm 2** SMP Inference (Sampling)

1: **Input**: orthonormal basis $B$; state-only router $p_\phi$; diffusion experts for $z$; budget $k$; coverage $\tau_m$
2: **while** task not complete **do**
3:     Observe $s_t$ and get router mean $\bar{g}_t = \mathbb{E}[g_t \mid s_t]$
4:     Set masses $m_i = \bar{g}_{t,i}^2$
5:     Initialize active set $S \leftarrow \varnothing$; $M_{\mathrm{tot}} = \sum_{j=1}^{K} m_j$
6:     **while** $|S| < k$ and $\sum_{i \in S} m_i / M_{\mathrm{tot}} < \tau_m$ **do**
7:         Define $F(S) = \sum_{i \in S} m_i$
8:         Choose $i^\star = \arg\max_{j \notin S} \left( F(S \cup \{j\}) - F(S) \right)$
9:         Update $S \leftarrow S \cup \{i^\star\}$,
10:     **end while**
11:     Denoise only $z_{t,S}$ (set $z_{t,j} = 0$ for $j \notin S$)
12:     Decode and execute $a_t = B(\bar{g}_t \odot z_t)$
13: **end while**

---

## 5 EXPERIMENTS

In this section, we evaluate SMP on multi-task bimanual manipulation using both simulation benchmarks and real-robot experiments. We investigate the following questions:

- Does SMP improve multi-task success while reducing inference cost compared to strong diffusion baselines?
- Do the *orthonormal skill basis* and *sticky routing* yield stable, phase-consistent behavior—with fewer gate switches/oscillations—and better cross-task skill reuse than unstructured mixtures?
- Can *adaptive expert activation* maintain reconstruction quality and success while substantially reducing active parameters and latency versus dense activation and FFN-MoE-styled methods [1]?
- After multi-task training, does the policy adapt to new tasks with limited demonstrations more effectively than baselines?

---

[1] FFN-MoE is mixture-of-experts only on an individual feed-forward neural network (Shazeer et al., 2017).

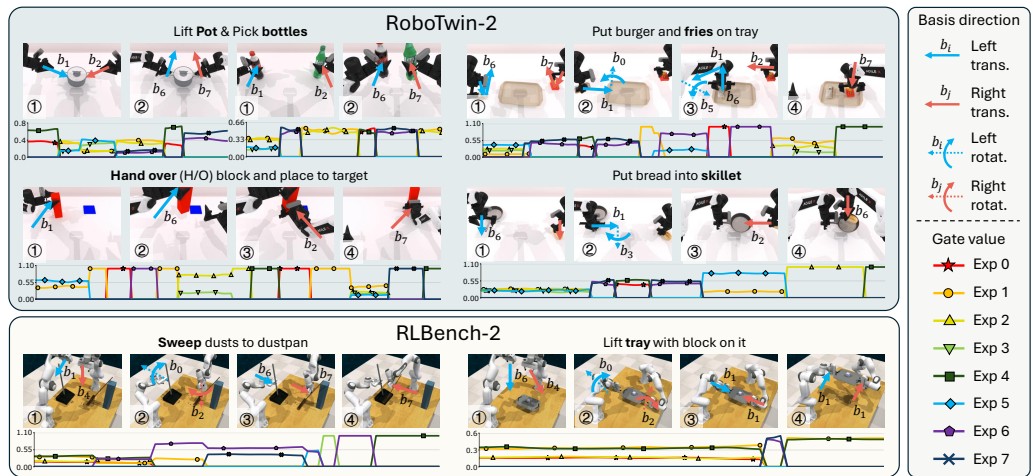

Figure 3: **Multi-task learning in RoboTwin-2 and RLBench-2.** SMP partitions bimanual control into an orthonormal skill basis and routes with sticky gates. Across tasks, the same experts are reused for left- and right-arm primitives and for pick–move–place phases, with few switches and long segments. Gate traces reveal sparse, phase-consistent activation, and cross-task skill reuse, indicating that actions are composed from a small, task-relevant subset of experts.

## 5.1 Experiment Setting

**Simulation Environments**. We evaluate on two bimanual manipulation benchmarks—RoboTwin-2 (Chen et al., 2025) and RLBench-2 (Grotz et al., 2024)—in which an agent controls two arms that may act independently or cooperatively. The enlarged action space and multimodal action distributions challenge conventional policies, while the natural decomposition into skill-specific subspaces enables cross-task skill reuse. We study *multi-task* learning and two forms of *transfer* on RoboTwin-2. In the **multi-task learning** setting, we train separate policies: one jointly on 6 RoboTwin-2 tasks (cross-arm skill reuse) and another jointly on 4 tightly coordinated RLBench-2 tasks (spatiotemporal collaboration). For transfer, we consider: (i) **few-shot adaptation**, where the RoboTwin-2 multi-task policy is fully fine-tuned on 4 new RoboTwin-2 tasks with 10 demonstrations each; and (ii) **skill composition**, where experts (and the skill basis) are frozen and only the router is fine-tuned on 2 new RoboTwin-2 tasks with 10 demonstrations each to test recomposition of learned skills. Additional environment details are provided in Appendix B.

**Baseline Methods and Evaluation Metrics**. Baselines include vanilla diffusion policies (DP (Chi et al., 2023), DP3 (Ze et al., 2024)), a transformer-based policy (ACT (Zhao et al., 2023)), a fine-tuned diffusion foundation model (RDT (Liu et al., 2024)), an information-theoretic skill-abstraction policy (Discrete Policy (Wu et al., 2025)), and an MoE diffusion policy with feed-forward experts (Sparse Diffusion Policy (Wang et al., 2024)). In each environment, policies are trained jointly across multiple tasks, increasing data complexity and stressing model capacity under multimodal distributions. We report task success rate, measures of learned skill abstraction, and computational cost and sampling efficiency. For real-world experiments, we report and compare the progress scores.

## 5.2 Multi-Task Learning Experimental Results

**Baseline methods struggle to generalize in bimanual multi-task settings.** Across both suites, DP, DP3, and ACT underfit the multimodal distributions induced by large bimanual action spaces and multiple tasks, yielding low success (Tab. 1). Scaling via RDT increases parameters by $10\times$ over DP (Tab. 2) but improves success by only $19\%$, indicating limited returns and the inefficiency of naïve model scaling for multimodal multi-task control. Discrete Policy separates some behaviors yet routes into a single backbone trained on heterogeneous data and quickly saturates without substantially larger capacity. Sparse Diffusion Policy replaces the backbone with FFN-MoE experts but lacks explicit geometric skill disentanglement; during sampling its gates switch frequently (higher flip-rate, introducing action oscillations and hurting precision-task success (Wang et al., 2024).

**SMP abstracts reusable manipulation skills across bimanual tasks and improves overall success.** As reported in Tab. 1 and visualized in Fig. 3, the policy partitions action generation across space and over time, routing to experts with consistent semantics. In RoboTwin-2 and RLBench-2,

Table 1: Success Rates in Bimanual Multi-Task Learning Tasks ↑

| Methods* | RobotTwin-2[†] | | | | | | | RLBench-2[†] | | | | |
|---|---|---|---|---|---|---|---|---|---|---|---|---|
| | Bottle | H/O | Pot | Fries | Skillet | Cab. | Avg. | Tray | Rope | Oven | Sweep | Avg. |
| DP | 0.21 | 0.17 | 0.34 | 0.54 | 0.11 | 0.37 | 0.29 | 0.10 | 0.13 | 0.08 | 0.09 | 0.10 |
| DP3 | 0.26 | 0.31 | 0.46 | 0.59 | 0.06 | 0.32 | 0.33 | - | - | - | - | - |
| ACT | 0.31 | 0.43 | 0.61 | 0.50 | 0.08 | 0.15 | 0.34 | 0.15 | 0.15 | 0.12 | 0.18 | 0.15 |
| RDT | 0.55 | 0.60 | 0.63 | 0.66 | 0.05 | 0.43 | 0.48 | 0.18 | 0.17 | 0.13 | **0.20** | 0.17 |
| Disc. Policy | 0.29 | 0.23 | 0.46 | 0.74 | 0.13 | **0.52** | 0.40 | 0.13 | 0.14 | 0.07 | 0.17 | 0.13 |
| Sparse DP | 0.37 | 0.42 | 0.59 | 0.77 | 0.08 | 0.42 | 0.44 | 0.15 | 0.18 | 0.12 | 0.17 | 0.16 |
| SMP (ours) | **0.56** | **0.61** | **0.64** | **0.79** | **0.14** | **0.52** | **0.54** | **0.19** | **0.20** | **0.14** | **0.20** | **0.18** |

[†] RoboTwin-2 (Chen et al., 2025) , RLBench-2 (Grotz et al., 2024). Tasks see Fig. 3. Results are averaged in 100 episodes. * DP (Chi et al., 2023), DP3 (Ze et al., 2024), ACT (Zhao et al., 2023), RDT (Liu et al., 2024), Sparse DP (Wang et al., 2024), Discrete Policy (Wu et al., 2025), SMP denotes our Skill Mixture-of-Experts Policy.

we observe two dominant patterns tied to our design: (i) left- and right-arm behaviors are handled by different experts—for example, one expert group consistently produces left-arm translation/rotation for grasping, while other experts govern the right arm's task-specific motions and (ii) trajectories organize into *pick* (pre-grasp, grasp), *move*, and *place* (pre-release, release) phases, where move/pre-release primarily invoke translation-focused experts, grasp/release are captured by gripper-focused experts, and pre-grasp combines translation and rotation experts for precise alignment. These structured patterns indicate reusable skills across tasks and align with the consistently higher (or comparable) success rates observed in the bimanual multi-task environments.

**SMP reduces active computation and latency via adaptive expert activation.** Tab. 2 reports total parameters $\mathcal{N}_p$, *active* parameters during sampling $\mathcal{N}_p^{\text{act}}$, and inference time $T_{\text{inf}}$. Rather than evaluating all experts at every step, SMP activates a small, state-dependent subset (top-$k$/coverage), so most timesteps use only a fraction of the backbone. In practice, it activates about $30\%$ of its own parameters—roughly $7\%$ of RDT's total—while maintaining high success. This selective routing yields consistently lower $T_{\text{inf}}$ than single-backbone diffusion baselines (DP, DP3, RDT, Discrete Policy) because only the chosen experts are denoised and composed in the final action, and multiple small experts can run in parallel. By contrast, the FFN-MoE baseline (used in Sparse Diffusion Policy) offers limited speedup since only

Table 2: Computation costs ↓

| Methods | $\mathcal{N}_p$ | $\mathcal{N}_p^{\text{act}}$ | $T_{\text{inf}}$ |
|---|---|---|---|
| DP | 132.5 | 132.5 | 120.3 |
| DP3 | 128.5 | 128.5 | 122.1 |
| ACT | 83.9 | 83.9 | 94.8 |
| RDT | 1200 | 1200 | 183.1 |
| Disc. Policy | 162.9 | 162.9 | 153.7 |
| Sparse DP | 154.4 | 110.1 | 148.3 |
| SMP (ours) | 258.9 | 80.2 | 107.3 |

$\mathcal{N}_p$: total network parameters (M), $\mathcal{N}_p^{\text{act}}$: expected active network parameters (M), $T_{\text{inf}}$: inference time (ms). Results are averaged over all tasks.

part of its computation is parallelizable and each denoising step must synchronize across experts, introducing overhead. We also observe the expected trade-off: increasing the activation budget improves reconstruction slightly but raises latency; our reported operating point is selected on validation to balance success and $T_{\text{inf}}$.

**Real-robot case study.** As shown in Fig. 5, we conducted a real-robot multi-task case-study on four bimanual manipulation tasks—*remove pen cap*, *put cups into bowl*, *hand over screwdriver*, and *pour beans into bowl* on the PiPER platform (AgileX, 2025). For each task, we collect 50 demonstrations and train a single multi-task policy per method. Performance is reported as the *progress score* (progress of task completion) averaged over 10 trials. Details in Appendix B.1.2.

Across all four tasks, *SMP* attains the highest average progress score over 10 trials while using fewer active parameters and lower inference latency than diffusion baselines (Fig. 5). RDT reaches competitive progress but incurs the largest compute and slowest inference on our setup. Discrete Policy and Sparse Diffusion Policy (SDP) operate at lower budgets yet trail SMP in progress, and their routing occasionally produces inconsistent phase selections. In contrast, SMP composes actions from a compact, task-relevant subset of experts with sticky routing, delivering steady progress and efficient execution.

## 5.3 TRANSFER LEARNING EXPERIMENTAL RESULTS

**Few-shot transfer.** We evaluated few-shot transfer on RoboTwin-2 by fully fine-tuning the multi-task policy on four new tasks with 10-shot adaptation. The results summarized in Tab. 3 show that DP and RDT transfer poorly; prior behaviors remain latent in large backbones and the limited data

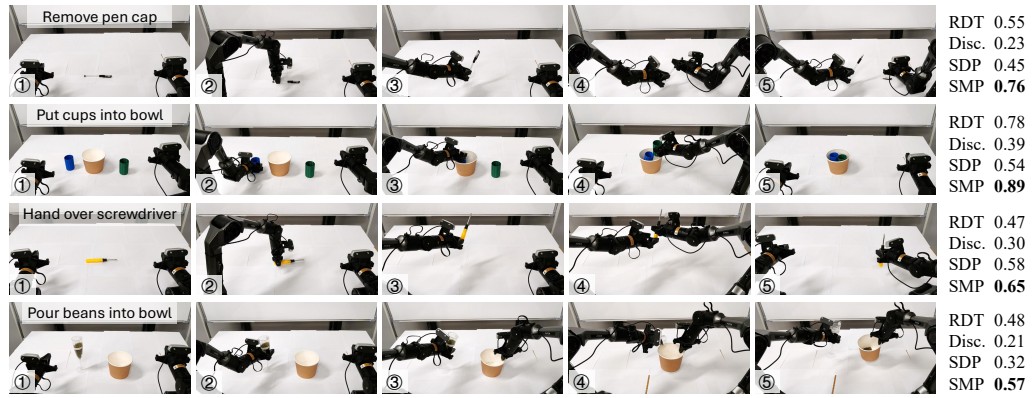

Figure 5: Real-robot experiments with four manipulation policies. **Left**: SMP executes 4 bimanual manipulation tasks. **Right**: Progress score ↑ of each task averaged in 10 trials.

was insufficient to obtain strong performance. Discrete Policy was able to reuse some latent codes, but the limited samples did not reliably tune its policy. In contrast, SMP applies MoE such that only the relevant experts are activated and fine-tuned without adding new experts; concentrating the ten-shot data on this sparse subset yielded the best success.

**Skill composition.** We study skill re-composition by transfer learning on two new RoboTwin-2 tasks that reuse left– and right–hand behaviors from the multi-task set (e.g., *fries → skillet* and *bottle → cabinet*; see Fig. 4). Each task provides 10 demonstrations; we freeze all experts (and the skill basis) and fine-tune only the router to test recombination without updating expert parameters.

SMP achieves higher performance than Sparse Diffusion Policy (SDP) (success rates reported in Tab. 4). SDP's layer-wise gating couples experts across diffusion steps, so router tuning did not isolate skills; qualitatively, we observed that per-arm roles blur—e.g., hesitant grasps in *fries→skillet* and misaligned placement in *bottle→cabinet*. In contrast, SMP encodes separates skills cleanly, enabling router-only updates to recompose right/left pick–and–place, which yielded steadier grasps and placements.

Table 3: Success Rate in Few-shot Transfer

| Methods | Div.[1] | Mic[2] | Roller[3] | Box[4] | Avg. |
|---|---|---|---|---|---|
| DP | 0.06 | 0.13 | 0.18 | 0.16 | 0.13 |
| RDT | 0.14 | 0.26 | 0.21 | 0.18 | 0.20 |
| Disc. Policy | 0.17 | 0.38 | 0.44 | 0.25 | 0.31 |
| SMP (ours) | **0.22** | **0.49** | **0.49** | **0.31** | **0.38** |

[1] Pick two diverse bottles. [2] Hand over a mic. [3] Grab a cooking roller. [4] Put two cans into a box. Results averaged over 100 episodes.

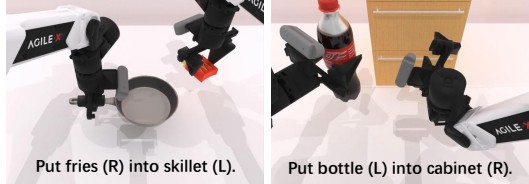

Put fries (R) into skillet (L).     Put bottle (L) into cabinet (R).

Figure 4: Skill Composition Tasks

Table 4: Success Rate in Skill Composition

| Methods | Skillet-Fries[1] | Bottle-Cab.[2] | Avg. |
|---|---|---|---|
| SDP | 0.11 | 0.38 | 0.25 |
| SMP (ours) | **0.15** | **0.44** | **0.30** |

[1] Pick fries (right) and place in skillet (left).
[2] Pick bottle (left) and place in cabinet drawer (right).

## 6 CONCLUSION

We introduced SMP, a diffusion-based mixture-of-experts policy that composes actions through a compact state-adaptive orthogonal skill basis with sticky routing and adaptive activation. By routing sparsely and stably in this basis, SMP yields phase-consistent skills that are reused across tasks; in simulation and on a real dual-arm platform, it delivers higher multi-task and transfer success at substantially lower inference cost than large diffusion baselines, enabling real-time control. Beyond success rates, the learned experts align with left/right-arm roles and pick–move–place phases, and the adaptive expert activation provides a simple, effective knob to trade off accuracy and latency.

**Limitations and future work.** This study uses relatively small diffusion backbones and focuses on bimanual manipulation; next we will scale SMP to larger models and datasets, broaden evaluation to single-arm and mobile manipulation, and run more extensive real-robot studies. We will also conduct targeted ablations of sticky routing and adaptive activation to quantify success–latency trade-offs and assess robustness under sensing noise and domain shift.

## ACKNOWLEDGEMENTS

This research / project is supported by the National Research Foundation, Singapore, under its Thematic Competitive Research Programme 2025 (NRF-T-CRP-2025-0003). The authors would also like to acknowledge support from the Google Foundation.

## ETHICS STATEMENT

This work adheres to the ICLR Code of Ethics and follows principles of responsible stewardship, avoidance of harm, fairness, and respect for privacy. Human operators provided teleoperation demonstrations but no user study/human-subject experiment was conducted and no personal data was collected.

## REPRODUCIBILITY STATEMENT

All algorithms, model architectures, and training procedures are described in the paper and Appendix A. Experimental details are provided in Appendix B. Hyperparameters, dataset sources, and pre-processing steps are specified. We also provide robot hardware details to support the replicability of experiments.

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

DISCLOSURE OF LLM USE

We used a large language model (LLM) mainly to check and correct grammar and language issues in the manuscript. We also used an LLM to validate ideas and check for errors. The authors take full responsibility for all content and affirm that all scientific ideas, analysis, and conclusions were conceived and written by the authors.

# A SKILL MIXTURE-OF-EXPERTS POLICY METHOD DETAILS

In Sec. 4, we presented the SMP formulation and its training and inference procedures. This appendix provides the accompanying mathematical and implementation details: (i) preliminaries on simplex-valued gates, Dirichlet–Markov (sticky) dynamics, and thin–QR retraction for the state-dependent basis; (ii) the subspace-decomposition assumption and the resulting variational objective for joint generation; (iii) the coefficient-space reconstruction targets and gate regularization terms; (iv) adaptive expert activation during inference; and (v) a breakdown of computation cost for training and sampling.

## A.1 MATHEMATICAL PRELIMINARIES

In this subsection, we collect the basic mathematical notions used in Section 4.1: simplex-valued gates, Dirichlet–Markov ("sticky") dynamics, and the thin–QR retraction with sign stabilization for constructing the orthonormal skill basis $B(s)$.

**Simplex weights.** We represent expert-selection weights as elements of the probability simplex

$$\Delta^{K-1} = \left\{ g \in \mathbb{R}_{\geq 0}^K \mid \mathbf{1}^\top g = 1 \right\}, \tag{9}$$

where $K$ is the number of skills and $\mathbf{1}$ is the all-ones vector. A vector $g \in \Delta^{K-1}$ can be interpreted as a categorical distribution over experts or as mixture weights in a convex combination of expert contributions. Throughout, we write $g_t \in \Delta^{K-1}$ for the gate at time $t$ in Eq. 1.

**Dirichlet distribution and Dirichlet–Markov dynamics.** The Dirichlet distribution is a standard prior over simplex-valued random variables. For a concentration parameter $\alpha \in \mathbb{R}_{>0}^K$, a random vector $g \sim \text{Dir}(\alpha)$ satisfies $g \in \Delta^{K-1}$ and has density

$$p(g \mid \alpha) = \frac{1}{Z(\alpha)} \prod_{k=1}^K g_k^{\alpha_k - 1}, \quad g \in \Delta^{K-1}, \tag{10}$$

where $Z(\alpha)$ is the normalizing constant. Larger $\alpha_k$ encourages $g_k$ to be larger on average, while smaller $\alpha_k$ pushes probability mass away from the $k$-th corner of the simplex.

To describe the temporal evolution of gates $g_{1:T}$, we use a first-order Markov process with Dirichlet conditionals, which we refer to as *Dirichlet–Markov dynamics*. In the main text (Eq. 3), this takes the form

$$\vartheta \sim \text{Dir}(\alpha\,\mathbf{1}), \quad g_1 \sim \text{Dir}(\alpha_0\,\vartheta), \quad g_t \sim \text{Dir}\big(\kappa\,g_{t-1} + \alpha_0\,\vartheta\big), \quad t \geq 2, \tag{11}$$

where $\vartheta \in \Delta^{K-1}$ is a global usage vector, $\alpha_0 > 0$ controls how strongly $\vartheta$ influences the gates, and $\kappa \geq 0$ controls the dependence on the previous gate $g_{t-1}$. Intuitively, the term $\alpha_0\,\vartheta$ encourages overall usage patterns that match $\vartheta$, while the term $\kappa\,g_{t-1}$ encourages the process to stay close to its previous value.

**Sticky Dirichlet–Markov dynamics.** In manipulation, we often expect *phase-like* behavior, where the active skills stay roughly constant over several timesteps. To capture this, we use *sticky* Dirichlet–Markov dynamics: in Eq. 11, a larger value of $\kappa$ increases the self-reinforcing effect of $g_{t-1}$, making the conditional distribution of $g_t$ more concentrated around the previous gate. In the limit of large $\kappa$, $g_t$ changes only slowly over time. This induces smooth, piecewise-constant gate trajectories while still allowing occasional switches between experts, and is exactly the prior used in the gate regularization term $\mathcal{L}_{\text{gate}}$ in Eq. 8.

**Thin–QR retraction with sign stabilization.** The skill basis $B(s) \in \mathbb{R}^{d \times K}$ in Eq. 1 is required to have orthonormal columns, i.e.,

$$B(s)^\top B(s) = I_K, \tag{12}$$

for each state $s$. The set of such matrices is the *Stiefel manifold*

$$\text{St}(d, K) = \left\{ B \in \mathbb{R}^{d \times K} \,\big|\, B^\top B = I_K \right\}. \tag{13}$$

Rather than optimizing $B(s)$ directly under this constraint, we maintain an unconstrained matrix $W(s) \in \mathbb{R}^{d \times K}$ (the output of a small neural network) and map it to $\text{St}(d, K)$ via a thin–QR factorization:

$$W(s) = \tilde{B}(s)\, U(s), \tag{14}$$

where $\tilde{B}(s) \in \mathbb{R}^{d \times K}$ has orthonormal columns and $U(s) \in \mathbb{R}^{K \times K}$ is upper triangular. A naive choice $B(s) = \tilde{B}(s)$ suffers from a sign ambiguity: multiplying a column of $\tilde{B}(s)$ by $-1$ and the corresponding row of $U(s)$ by $-1$ leaves $W(s)$ unchanged, but flips the direction of the basis vector.

To avoid such discontinuous flips as $s$ varies, we apply *sign stabilization*. Let

$$D(s) = \text{diag}\big(\text{sign}(\text{diag}(U(s)))\big), \tag{15}$$

and define

$$B(s) = \tilde{B}(s)\, D(s). \tag{16}$$

This enforces a consistent sign convention (e.g., making the diagonal entries of $U(s)$ positive) and yields an orthonormal basis $B(s)$ that evolves smoothly with the state. In practice, we treat $W(s)$ as the learnable parameter, apply the QR-based map $W(s) \mapsto B(s)$ in each forward pass, and backpropagate through this retraction using automatic differentiation. This implements the state-adaptive orthogonal frame described in Eq. 2 and used throughout Section 4.

## A.2 Subspace decomposition and variational inference

The decoder in Eq. 1 assumes that, at each state $s_t$, the action $a_t \in \mathbb{R}^d$ can be synthesized from a $K$-dimensional skill subspace spanned by the columns of the orthonormal basis $B(s_t)$:

$$a_t \approx B(s_t)\big(g_t \odot z_t\big), \qquad B(s_t)^\top B(s_t) = I_K, \tag{17}$$

where $K \ll d$ and $g_t \in \Delta^{K-1}$, $z_t \in \mathbb{R}^K$ are the gate and skill coefficients. We first make explicit the geometric assumption underlying this decomposition and how possible *remainders* are handled, and then derive the resulting variational formulation.

**Subspace decomposition and remainder.** In full generality, any $a_t \in \mathbb{R}^d$ can be decomposed into a part lying in the skill subspace and an orthogonal remainder. Let $B(s_t) \in \mathbb{R}^{d \times K}$ be an orthonormal basis and $R(s_t) \in \mathbb{R}^{d \times (d-K)}$ an orthonormal complement satisfying

$$B(s_t)^\top B(s_t) = I_K, \quad R(s_t)^\top R(s_t) = I_{d-K}, \quad B(s_t)^\top R(s_t) = 0. \tag{18}$$

Then every action admits an exact decomposition

$$a_t = B(s_t)\, u_t + R(s_t)\, r_t, \tag{19}$$

for some coefficients $u_t \in \mathbb{R}^K$ and $r_t \in \mathbb{R}^{d-K}$. In our model we *identify* the structured component $u_t$ with the gated skill coefficients,

$$u_t = g_t \odot z_t, \tag{20}$$

and treat the remainder $R(s_t) r_t$ as a small residual. Rather than modeling $r_t$ explicitly, we absorb it into a small-variance Gaussian likelihood around the reconstructed action:

$$p\big(a_t \mid g_t, z_t, s_t, B\big) = \mathcal{N}\big(B(s_t)(g_t \odot z_t), \sigma_a^2 I\big), \qquad \sigma_a^2 \ll 1. \tag{21}$$

Thus, in Eq. 1 we explicitly model only the low-dimensional, task-relevant component $B(s_t)(g_t \odot z_t)$ and regard any energy outside $\text{span}\{B(s_t)\}$ as reconstruction noise.

The key modeling assumption is that, for a suitable choice of $K$, the demonstration actions lie *approximately* in a $K$-dimensional state-dependent subspace:

$$\|R(s_t) r_t\|_2 \text{ is small for all } t, \tag{22}$$

so that the residual term can be neglected without materially affecting the policy. This is analogous to low-rank models (e.g., PCA), where most of the variance is captured by a small number of principal directions, while the remaining directions contribute only minor corrections. By constraining the decoder to $B(s_t)$, we deliberately force actions to be expressed through a small set of reusable skill directions, which improves identifiability of skills, stabilizes gating, and makes coefficient supervision in the whitened basis well-conditioned. In practice, we found that choosing a modest $K$ suffices to reconstruct manipulation actions accurately, so we omit an explicit residual branch in the main formulation and let the Gaussian likelihood account for any small remainder.

**Variational inference for joint generation.** Given states $s_{1:T}$ and actions $a_{1:T}$, SkillMoE synthesizes each action in this state-dependent orthonormal skill subspace,

$$a_t = B(s_t)(g_t \odot z_t), \qquad B(s_t)^\top B(s_t) = I_K, \quad g_t \in \Delta^{K-1}, \; z_t \in \mathbb{R}^K, \tag{23}$$

and we write $B = B(s_t)$ below for brevity. The likelihood is given by the Gaussian model above, and we place a sticky Dirichlet prior on the gates with a global-usage variable $\vartheta$ (Eq. 3); coefficient priors follow the diffusion construction:

$$p(a_t \mid g_t, z_t, s_t, B) = \mathcal{N}(B(g_t \odot z_t), \sigma_a^2 I), \qquad \sigma_a^2 \ll 1, \tag{24}$$

$$p(\vartheta, \mathbf{g} \mid s_{1:T}) = p(\vartheta)\, p(g_1 \mid \vartheta) \prod_{t=2}^{T} p(g_t \mid g_{t-1}, \vartheta), \tag{25}$$

$$p(\mathbf{z} \mid s_{1:T}) = \prod_{t=1}^{T} p(z_t \mid s_t), \tag{26}$$

where $\mathbf{g} = g_{1:T}$ and $\mathbf{z} = z_{1:T}$.

The trajectory joint conditioned on $s_{1:T}$ is

$$p_\theta(\vartheta, \mathbf{g}, \mathbf{z}, a_{1:T} \mid s_{1:T}) = \Big[ \prod_{t=1}^{T} p(a_t \mid g_t, z_t, s_t, B) \Big] p(\vartheta, \mathbf{g} \mid s_{1:T})\, p(\mathbf{z} \mid s_{1:T}). \tag{27}$$

We adopt a factorized amortized posterior

$$q(\vartheta, \mathbf{g}, \mathbf{z} \mid a_{1:T}, s_{1:T}) = q(\vartheta, \mathbf{g} \mid a_{1:T}, s_{1:T})\, q(\mathbf{z} \mid a_{1:T}, s_{1:T}), \tag{28}$$

with Dirichlet amortizers for $\vartheta$ and $g_t$, and diffusion posteriors for $z_t$. By Jensen's inequality,

$$\log p_\theta(a_{1:T} \mid s_{1:T}) = \log \int p_\theta(\vartheta, \mathbf{g}, \mathbf{z}, a_{1:T} \mid s_{1:T})\, d\vartheta\, d\mathbf{g}\, d\mathbf{z} \tag{29}$$

$$= \log \int q(\vartheta, \mathbf{g}, \mathbf{z} \mid a_{1:T}, s_{1:T}) \frac{p_\theta(\vartheta, \mathbf{g}, \mathbf{z}, a_{1:T} \mid s_{1:T})}{q(\vartheta, \mathbf{g}, \mathbf{z} \mid a_{1:T}, s_{1:T})}\, d\vartheta\, d\mathbf{g}\, d\mathbf{z} \tag{30}$$

$$\geq \mathbb{E}_q \Bigg[ \sum_{t=1}^{T} \log p(a_t \mid g_t, z_t, s_t, B) + \log p(\vartheta, \mathbf{g} \mid s_{1:T}) + \log p(\mathbf{z} \mid s_{1:T}) \tag{31}$$

$$- \log q(\vartheta, \mathbf{g} \mid a_{1:T}, s_{1:T}) - \log q(\mathbf{z} \mid a_{1:T}, s_{1:T}) \Bigg]. \tag{32}$$

Collecting terms,

$$\mathcal{L}_{\text{ELBO}} = \underbrace{\mathbb{E}_q \Big[ \sum_{t=1}^{T} \log p(a_t \mid g_t, z_t, s_t, B) \Big]}_{\mathcal{L}_{\text{recon}}} - \underbrace{D_{\text{KL}}\Big( q(\vartheta, \mathbf{g} \mid a_{1:T}, s_{1:T}) \,\big\|\, p(\vartheta, \mathbf{g} \mid s_{1:T}) \Big)}_{\mathcal{L}_{\text{gate}}} \tag{33}$$

$$- \underbrace{D_{\text{KL}}\Big( q(\mathbf{z} \mid a_{1:T}, s_{1:T}) \,\big\|\, p(\mathbf{z} \mid s_{1:T}) \Big)}_{\mathcal{L}_{\text{coeff}}}.$$

**Implementation notes.** (i) The Gaussian likelihood yields an MSE in the whitened space that directly supervises the (state-dependent) basis $B$ (Sec. A.3). (ii) The gate KL decomposes into global-usage, initial-gate, and sticky terms (Sec. A.4). (iii) The coefficient KL is realized by the standard DDPM surrogate on $z_t$ using clean targets formed by projecting $a_t$ into the basis (Sec. A.3). Only $\mathcal{L}_{\text{recon}}$ backpropagates through $B$; $\mathcal{L}_{\text{coeff}}$ does not.

A.3 RECONSTRUCTION TARGET

To stabilize training while ensuring the (state-dependent) basis $B(s_t)$ captures action variability, we form *two* coefficient targets from the same projection, differing only in whether gradients flow to $B$:

$$\hat{z}_{0,t}^{\text{sg}} = \frac{\bar{B}(s_t)^\top a_t}{\mathbb{E}_q[g_t] + \epsilon}, \qquad \hat{z}_{0,t}^{\text{rec}} = \frac{B(s_t)^\top a_t}{\mathbb{E}_q[g_t] + \epsilon}, \qquad \bar{B}(s_t) = \text{sg}[\,B(s_t)\,], \ \ \epsilon \approx 10^{-3}, \qquad (34)$$

where the division is elementwise and $\mathbb{E}_q[g_t]$ is the mean of the amortized gate posterior. The stop–gradient copy $\bar{B}(\cdot)$ prevents the diffusion loss from directly changing the basis, while the reconstruction path uses a gradient-carrying projection so that $B(\cdot)$ is updated toward capturing action energy.

Using the gradient-carrying target, we decode

$$\hat{a}_t^{\text{rec}} \ = \ B(s_t)\big(g_t \odot \hat{z}_{0,t}^{\text{rec}}\big), \qquad \mathcal{L}_{\text{recon}} \ = \ \frac{1}{2\sigma_a^2} \sum_{t=1}^{T} \big\|a_t - \hat{a}_t^{\text{rec}}\big\|_2^2. \qquad (35)$$

In parallel, we train the coefficient denoisers by re-noising the *stop–gradient* clean targets $\hat{z}_{0,t}^{\text{sg}}$ and applying the standard DDPM objective (omitting constants),

$$\mathcal{L}_{\text{coeff}} \ = \ \sum_{t=1}^{T} \mathbb{E}_{\tau,\varepsilon}\Big[\big\|\varepsilon - \varepsilon_\psi(z_t^{(\tau)}, \tau, s_t)\big\|_2^2\Big], \qquad z_t^{(\tau)} \ = \ \sqrt{\bar{\alpha}_\tau}\,\hat{z}_{0,t}^{\text{sg}} + \sqrt{1 - \bar{\alpha}_\tau}\,\varepsilon, \ \ \varepsilon \sim \mathcal{N}(0, I).$$
$$(36)$$

Crucially, gradients from $\mathcal{L}_{\text{coeff}}$ do not affect $B(\cdot)$ (they flow only to the denoisers and to the amortizers that produced $\mathbb{E}_q[g_t]$), while $\mathcal{L}_{\text{recon}}$ updates $B(\cdot)$ (and $g_t$) through both the projection $B(s_t)^\top a_t$ and the decoder $B(s_t)(\cdot)$.

Projecting with $B(s_t)^\top$ concentrates action energy into $K$ orthogonal directions; training with $\hat{z}_{0,t}^{\text{rec}}$ makes the whitened decoder $B(s_t)\big(g_t \odot z_t\big)$ sufficient to match $a_t$, pushing any unexplained variation toward zero. In effect, $B(\cdot)$ is encouraged to span the task-relevant subspace (skills), and reconstruction error vanishes when the basis captures all structure. This yields compact, interpretable skills and stabilizes routing, since non-overlapping directions reduce ambiguity in $g_t$.

A.4 STICKY GATE REGULARIZATION

We parameterize the gate dynamics with a global usage vector and a first-order *sticky* process:

$$\vartheta \sim \text{Dir}(\alpha\,\mathbf{1}), \qquad g_1 \sim \text{Dir}(\alpha_0\,\vartheta), \qquad g_t \sim \text{Dir}\big(\kappa\,g_{t-1} + \alpha_0\,\vartheta\big), \ \ t \geq 2, \qquad (37)$$

with $\alpha > 0$ (diffuseness of global usage), $\alpha_0 > 0$ (anchor strength toward $\vartheta$), and $\kappa > 0$ (temporal stickiness). Increasing $\kappa$ lengthens segments (fewer switches), larger $\alpha$ balances usage across experts, and $\alpha_0$ prevents collapse.

Let $q(\vartheta) = \text{Dir}(\hat{\alpha})$ and $q(g_t) = \text{Dir}(\hat{\beta}_t)$ denote the amortized posteriors. The gate regularizer in the ELBO decomposes as

$$\mathcal{L}_{\text{gate}} = D_{\text{KL}}\big(q(\vartheta) \,\|\, \text{Dir}(\alpha\mathbf{1})\big) \ + \ D_{\text{KL}}\big(q(g_1) \,\|\, \text{Dir}(\alpha_0\,\mathbb{E}_q[\vartheta])\big)$$
$$+ \ \sum_{t=2}^{T} D_{\text{KL}}\Big(q(g_t) \,\Big\|\, \text{Dir}\big(\kappa\,\mathbb{E}_q[g_{t-1}] + \alpha_0\,\mathbb{E}_q[\vartheta]\big)\Big), \qquad (38)$$

encouraging (i) plausible global usage, (ii) a non-degenerate initial gate, and (iii) temporal persistence around $g_{t-1}$ with a soft pull toward $\vartheta$. We use the closed-form KL for Dirichlet distributions:

$$D_{\text{KL}}\big(\text{Dir}(\hat{\beta}) \,\|\, \text{Dir}(\beta)\big) = \log \frac{\Gamma(\sum_i \hat{\beta}_i)}{\Gamma(\sum_i \beta_i)} - \sum_i \log \frac{\Gamma(\hat{\beta}_i)}{\Gamma(\beta_i)} + \sum_i (\hat{\beta}_i - \beta_i)\Big(\psi(\hat{\beta}_i) - \psi\big(\textstyle\sum_j \hat{\beta}_j\big)\Big),$$
$$(39)$$

where $\psi(\cdot)$ is the digamma function.

At deployment, we use a *state-only* router $p_\phi(g_t \mid s_t) = \text{Dir}(\tilde{\beta}_\phi(s_t))$. To match the training-time amortized gates $q(g_t \mid s_t, a_t)$, we add an auxiliary alignment loss

$$\mathcal{L}_{\text{align}} = \sum_{t=1}^{T} D_{\text{KL}}\big(q(g_t \mid s_t, a_t) \,\|\, \text{Dir}(\tilde{\beta}_\phi(s_t))\big), \tag{40}$$

which improves test-time consistency and stabilizes sticky routing when actions are unavailable.

**Practical implementation.** We find it helpful to (i) anneal $\kappa$ from a small value to its target to avoid early over-stickiness; (ii) warm-start $\alpha_0$ to prevent expert collapse; (iii) optionally temperature-sharpen $\hat{\beta}_t$ during alignment to encourage sparse, phase-consistent gates; and (iv) monitor a *flip-rate* (fraction of steps with $\arg\max g_t \neq \arg\max g_{t-1}$) and *segment length* as diagnostics of temporal smoothness.

### A.5 ADAPTIVE EXPERT ACTIVATION

At test time, evaluating all $K$ experts at every step is unnecessary. We seek a *sparse*, state-dependent active set $S_t \subseteq \{1, \ldots, K\}$ that preserves policy quality while reducing latency. Using the *state-only* router $p_\phi(g_t \mid s_t) = \text{Dir}(\tilde{\beta}_\phi(s_t))$, we take its mean $\bar{g}_t = \mathbb{E}[g_t \mid s_t]$ and define the *mass* of expert $i$ as

$$m_i = \bar{g}_{t,i}^2.$$

Given an orthonormal skill basis $B = [b_1, \ldots, b_K]$ with $B^\top B = I$, the additive score

$$F(S) = \sum_{i \in S} m_i \tag{41}$$

is *modular* (additive). Therefore, sorting experts by $m_i$ is optimal for two selection regimes: (i) **top-$k$**, which chooses the $k$ largest-$m_i$ experts, and (ii) **coverage**, which chooses the smallest prefix of the sorted list whose cumulative mass reaches a target fraction $\tau_m$ of the total, i.e., $\sum_{i \in S} m_i / \sum_{j=1}^{K} m_j \geq \tau_m$ (we use $\tau_m \in [0.9, 0.95]$). After forming $S_t$, we denoise only $z_{t,S_t}$ (others set to zero) and decode $a_t = B(\bar{g}_t \odot z_t)$, yielding state-wise sparsity and reduced inference cost.

**Optimality and complexity.** For the top-$k$ constraint $\max_{|S| \leq k} \sum_{i \in S} m_i$, selecting the $k$ largest masses is *exactly optimal* (by additivity). For the coverage rule, the smallest prefix that attains the threshold $\tau_m$ is also *exactly optimal* for minimizing $|S|$ under the modular objective. The per-step complexity is $O(K \log K)$ for sorting (or $O(K)$ with partial selection when $k$ is small).

**Practical notes.** We cap $|S_t|$ by a small $k$ (e.g., 2–4) and set $\tau_m \in [0.9, 0.95]$ to stabilize latency with negligible accuracy loss. Mass-only selection requires no task metrics or Jacobians, is robust across embodiments, and integrates cleanly with sticky routing.

## B EXPERIMENTS DETAILS

### B.1 ENVIRONMENT SETTINGS

We evaluate multitask learning in both simulated and real-world settings. In simulation, we additionally conduct transfer-learning experiments to assess skill reusability and adaptation. Tasks are deliberately chosen or designed to stress close bimanual coordination, enabling a thorough and informative evaluation. Detailed task definitions and protocols are provided in Secs. B.1.1 and B.1.2.

#### B.1.1 SIMULATED EXPERIMENTS

**Multitask Learning.** For multitask learning experiments, we conduct experiments on 6 tasks from RoboTwin-2 (Chen et al., 2025) and 4 tasks from RLBench-2 (Grotz et al., 2024), separately. Methods are trained for 3000 epochs on Robotwin-2 and 300k iterations on RLBench-2.

Table 5: Selected bimanual tasks from **Robotwin-2** (Chen et al., 2025), **RLBench-2** (Grotz et al., 2024), and real-world evaluations. Each entry lists the task abbreviation, full name, and a brief description.

| Abbr. | Full name | Description |
|---|---|---|
| **RoboTwin-2** (Chen et al., 2025) (Multitask Learning) | | |
| Bottle | *Pick Dual Bottles* | Pick up one bottle with one arm, and pick up another bottle with the other arm. |
| H/O | *Handover Block* | Use the left arm to grasp the red block on the table, handover it to the right arm and place it on the blue pad. |
| Pot | *Lift Pot* | Use both arms to lift the pot. |
| Burger | *Place Burger Fries* | Use dual arm to pick the hamburg and frenchfries and put them onto the tray. |
| Skillet | *Place Bread Skillet* | There is one bread on the table. Use one arm to grab the bread and the other arm to take the skillet. Put the bread into the skillet. |
| Cab. | *Put Object Cabinet* | Use one arm to open the cabinet's drawer, and use another arm to put the object on the table to the drawer. |
| **RoboTwin-2** (Transfer Learning) | | |
| Div. | *Pick Diverse Bottles* | Pick up one bottle with one arm, and pick up another bottle with the other arm. |
| Mic. | *Handover Mic* | Use one arm to grasp the microphone on the table and handover it to the other arm. |
| Roller | *Grab Roller* | Use both arms to grab the roller on the table. |
| Box | *Place Cans Plasticbox* | Use dual arm to pick and place cans into plasticbox. |
| **RLBench-2** (Grotz et al., 2024) | | |
| Tray | *Lift Tray* | The robot's task is to lift a tray that is placed on a holder. An item is on top of the tray and must be balanced while both arms lift the tray. |
| Rope | *Straighten Rope* | The robot's task is to straighten a rope by manipulating it so that both ends are placed into distinct target areas. |
| Oven | *Take Tray Out of Oven* | The robot's task is to remove a tray that is located inside an oven. This involves opening the oven door and then grasping the tray. |
| Sweep | *Sweep Dust Pan* | The robot's task is to sweep the dust into the dust pan using a broom. This involves coordinating the sweeping motion to ensure the dust is effectively collected. |
| **Real-World Tasks** | | |
| / | *Remove pen cap* | On pen is placed on the table. Use on arm to pick the pen and the other arm to remove the cap. |
| / | *Put cups in bowl* | Two cups are placed on the table. Sequentially pick them and place them into the bowl. |
| / | *Handover screwdriver* | A tool is placed on the table. Use one arm to pick the tool and handover it to the other arm. |
| / | *Pouring Beans* | One bowl and a cup of beans are placed on the table. Use one arm to grab the cup and the other arm to grab the bowl. Then pour the beans into the bowl. |

We select tasks prioritizing scenarios that demand tight, real-time cooperation between the two arms (e.g., coordinated grasp-and-handover, cooperative manipulation under constraints, dual-arm assembly). Focusing on such challenging, interdependent tasks intentionally enlarges the skill base: policies must discover and reuse shared coordination primitives—spatial alignment, contact-rich synergies, role switching, and timing—across tasks, which in turn improves generalization and overall performance.

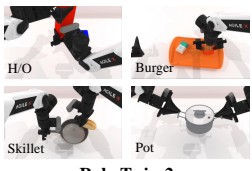 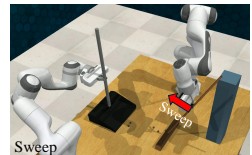 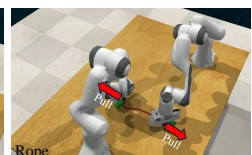 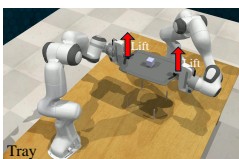

**RoboTwin-2**      **RLBench 2**

Figure 6: Examples of Simulated Experiments

**Transfer Learning.** For transfer-learning study, we select 4 bimanual tasks from RoboTwin-2. We first pre-train a single policy on the six tasks used in the multitask setting, then fine-tune that policy on each target task separately. The four targets are chosen to be skill-adjacent to the multi-task set—requiring similar coordination primitives while introducing new features such as different object geometries/materials, spatial layouts, or action sequences. This design directly tests each method's ability to transfer by reusing previously acquired skills, evaluating both zero-shot generalization and fine-tuning efficiency alongside final performance.

To further assess skill decomposition and cross-task re-composition, we construct two composite tasks on top of RoboTwin-2: *Put Fries into Skillet* and *Put Bottle into Cabinet*. Both are recombinations of base tasks used in our multitask training—*Put Fries into Skillet* combines *Skillet* and *Fries*, while *Put Bottle into Cabinet* combines *Bottle* and *Cabinet*. During training on these composites, we freeze the learned experts and bases and only finetune the gating network, which enables a clean evaluation of each method's ability to decompose and reuse skills.

In the main part, their full names are abbreviated for brevity; Tab. 5 lists the abbreviation–full-name mapping, brief descriptions, and the required level of bimanual coordination.

### B.1.2 REAL-WORLD EXPERIMENTS

To broaden evaluation beyond simulation and stress-test robustness under sensing noise, calibration drift, and actuation latency, we design and execute a suite of 4 real-world bimanual tasks— including *remove pen cap*, *put cups into bowl*, *hand over screwdriver*, and *pour beans into bowl*. These tasks demand tight inter-arm cooperation (role allocation, handoff timing, compliant contact) and precise motion control, making them deliberately challenging.

For each task, we collect 50 expert demonstration trajectories via human teleoperation. Training and evaluation follow a multitask regime: methods are trained for 2000 epochs on the union of demonstrations, with an explicit task identifier provided to the policy. All experiments are conducted on a platform of 2 PiPER arms. Detailed descriptions are provided in Tab. 5. The progress score reported in Fig. 5 is calculated based on the progress of task completion. And the progresses are defined as follows.

**Remove pen cap**

1. Progress 1/4: Left hand grasps pen.
2. Progress 2/4: Left hand lifts up pen.
3. Progress 3/4: Right hand grasps pen cap.
4. Progress 4/4: Pen cap is removed.

**Put cups in bowl**

1. Progress 1/4: Grasp and lift the left cup.
2. Progress 2/4: Place the left cup in the bowl.
3. Progress 3/4: Grasp and lift the right cup.
4. Progress 4/4: Place the right cup in the bowl.

**Handover screwdriver**

1. Progress 1/4: Left hand grasps the screwdriver.
2. Progress 2/4: Left hand lifts up the screwdriver.

3. Progress 3/4: Right hand grasps the screwdriver.

4. Progress 4/4: Left hand releases the screwdriver.

**Pouring beans**

1. Progress 1/5: Left hand grasps the cup with beans.

2. Progress 2/5: Left hand lifts up the cup and moves to the central area.

3. Progress 3/5: Right hand grasps the bowl.

4. Progress 4/5: Right hand lifts up the bowl and moves to the central area.

5. Progress 5/5: Left hand pours the beans into the bowl.

## B.2 BASELINES

We select strong, representative baselines to ensure persuasive comparisons. To use them in our multi-task bimanual setting, we apply only minimal adaptations while preserving each method's core design.

**DP and DP3.** Diffusion Policy (DP) (Chi et al., 2023): a conditional diffusion visuomotor policy that predicts action sequences from observations; a strong supervised IL baseline. 3D Diffusion Policy (DP3) (Ze et al., 2024): DP extended with 3D/SE(3)-aware modeling from point clouds, representative for precise spatial reasoning.

**ACT.** Action Chunking with Transformers (ACT) (Zhao et al., 2023): learns variable-length action primitives ("chunks") with a transformer policy—representative temporal-abstraction baseline for long-horizon control.

**RDT.** Robotics Diffusion Transformer (RDT) (Liu et al., 2024): a diffusion–transformer architecture for sequence-level action generation with a large size of 1.2B; representative modern baseline for multitask robot manipulation foundation model.

**Discrete Policy.** Discrete Policy (Wu et al., 2025) is an *info-based* skill-abstraction method that learns a discrete latent *codebook* of action "skills" via VQ-VAE and uses a conditional latent diffusion model to generate task-specific codes; the discrete bottleneck encourages skill disentanglement and makes DP a representative multi-task baseline (including bimanual settings).

**SDP.** Sparse Diffusion Policy (SDP) (Wang et al., 2024) is a representative MoE-based multitask imitation-learning method that performs skill abstraction and reuse via sparse gating. Concretely, it inserts layer-wise FFN experts inside the diffusion backbone and activates a small subset per step. Unlike our SMP, which *decomposes the entire policy* into separate experts (each a standalone policy/generator over a skill-specific subspace), SDP applies MoE only within network blocks—i.e., the final action is produced by a single diffusion head rather than an ensemble of independent policies.

## B.3 SMP IMPLEMENTATION DETAILS

**Optimization.** We optimize all models with AdamW (Loshchilov & Hutter, 2019), using a learning rate of $1 \times 10^{-5}$. The batch size is 128 for **Robotwin-2** and real-world experiments, and 6 for **RLBench-2**. We use an observation horizon of 3 steps and a planning horizon of 8 action steps. The SMP employs $K = 8$ experts with an activation threshold $\tau_m = 0.95$.

**Architecture.** SMP comprises: (i) a shared observation encoder; (ii) posterior and prior gating networks, $q(g_t \mid s_t, a_t)$ and $p(g_t \mid s_t)$, that produce expert weights; (iii) a basis generator $W(s_t)$; and (iv) $K$ per-expert diffusion generators that output coefficients $z_i$. At inference, experts with cumulative posterior/prior weight above $\tau_m$ are activated (with a top-1 fallback if none exceed the threshold), and their outputs are combined via the learned basis.

**Components.** *Observation encoder:* Following the standard observation encoder used in diffusion-based visuomotor policies (e.g., Diffusion Policy Chi et al. (2023)), we use a ResNet-18 that processes the RGB inputs together with the robot state and outputs a shared feature vector $s_t$, which is then used as input to the gating network, the state-adaptive skill-basis network, and the

diffusion experts. *Gating and basis:* both $q(g_t \mid s_t, a_t)$, $p(g_t \mid s_t)$, and $W(s_t)$ are implemented as lightweight MLPs for simplicity and speed; $W(s_t)$ maps the state to a task-adaptive set of bases used to synthesize the final action from expert coefficients. *Experts:* each expert is a diffusion generator following the CNN architecture of Diffusion Policy (Chi et al., 2023), with reduced channel widths to control model size.

**Computation cost.** All SMP models are trained end-to-end on a single NVIDIA A6000 GPU. For the multi-task experiments on RoboTwin-2.0 and RLBench-2, we train for 3000 epochs, which corresponds to roughly 20–25 hours of wall-clock time. The state-dependent orthonormal skill basis is learned jointly with the gating network and diffusion experts; no additional pre-training or separate optimization stage is required beyond this standard training procedure.

To make the sampling cost more transparent, Table 6 reports the number of parameters and the measured per-step inference time of each main component on the same A6000 GPU:

Table 6: Parameter count and per-step inference time of each SMP component.

| Module | # Parameters | Inference time |
|---|---|---|
| Vision encoder | 11.2M | 23 ms |
| Gate | 3.6M | 10 ms |
| Skill-basis network | 12.5M | 24 ms |
| Diffusion expert (single) | 28.9M | 74 ms |

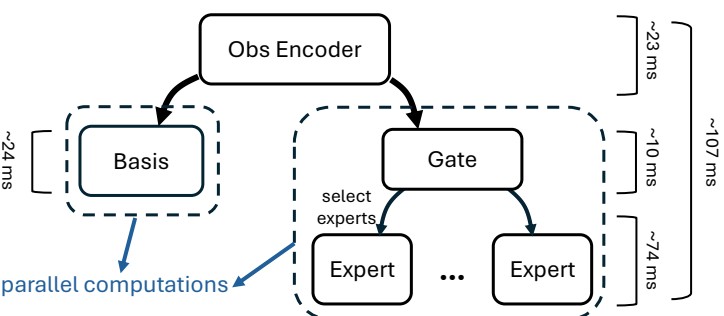

Figure 7: Inference pipeline of SMP. The observation encoder first maps the input observation to a shared feature, which is then fed to the state-dependent skill-basis network and to the MoE module (gate and selected diffusion experts). The basis and experts are evaluated in parallel, with the gate adding only a small overhead. The annotated values indicate the measured per-step runtimes of each component on an NVIDIA A6000 GPU, which sum to an overall inference time of approximately 107 ms per control step.

As illustrated in Fig. 7, the observation encoder is evaluated once per control step and its output feature is shared by all subsequent modules. Conditioned on this feature, the skill-basis network and the diffusion experts are evaluated in parallel, while the gating network selects the active experts with a relatively small additional cost. The resulting end-to-end inference time of SMP is approximately 107 ms per control step, and the overhead introduced by learning and using the state-dependent skill basis is moderate compared to the diffusion experts.

## C  ABLATION STUDIES

### C.1  ABLATION OF STICKY GATE FUNCTION

We briefly recall that SMP regularizes the skill gates $g_t \in \Delta^{K-1}$ with a sticky Dirichlet–Markov prior

$$\vartheta \sim \mathrm{Dir}(\alpha \, \mathbf{1}), \qquad g_1 \sim \mathrm{Dir}(\alpha_0 \, \vartheta), \qquad g_t \sim \mathrm{Dir}\big(\kappa \, g_{t-1} + \alpha_0 \, \vartheta\big), \ \ t \geq 2, \qquad (42)$$

Table 7: Success Rates of Ablation Studies ↑

| Methods | Multi-task Learning | | Transfer Learning | |
|---|---|---|---|---|
| | RoboTwin-2 | RLBench-2 | Few-shot Learning | Skill Composition |
| SMP (ours) | 0.54 | 0.18 | 0.38 | 0.30 |
| W/o sticky gate | 0.44 | 0.15 | 0.33 | 0.26 |
| W/o adaptive expert[1] | 0.53 | 0.18 | 0.37 | 0.29 |
| Linear mass adapt.[2] | 0.52 | 0.17 | 0.37 | 0.29 |
| Fixed skill basis | 0.40 | 0.14 | 0.31 | 0.24 |
| PCA skill basis[3] | 0.32 | 0.11 | 0.26 | 0.20 |

Results are averaged over all tasks. [1] Use top-k = 4 as default. [2] Change the mass definition of in adaptive expert activation as a linear function $m_i = \bar{g}_{t,i}$. [3] Directly use PCA to abstract the action space as fixed basis.

and train with closed-form Dirichlet KLs (Eq. 8). Intuitively, $\alpha$ shapes the global usage vector $\vartheta$, $\alpha_0$ controls how strongly each gate is pulled toward $\vartheta$, and $\kappa$ controls the temporal stickiness of $g_t$ around $g_{t-1}$. Unless otherwise stated we use $(\alpha, \alpha_0, \kappa) = (2.0, 0.5, 20.0)$.

To assess the importance of the sticky prior itself, we first remove it entirely and train a variant where a feedforward network predicts $g_t$ at each timestep without any Dirichlet–Markov structure or global-usage variable (no KL terms on $\vartheta$ or $g_{t-1}$). As reported in Table 7 (row "W/o sticky gate"), this leads to a clear degradation in performance: on RoboTwin-2 multi-task learning, the success rate drops from $0.54$ (SMP) to $0.44$, with similar declines on RLBench-2, few-shot transfer, and skill composition. Qualitatively, this model exhibits high-frequency switching between experts and less interpretable gate trajectories, indicating that temporal stickiness and global usage regularization are both important.

We then study the effect of each hyperparameter in Eq. 42. For each of $\alpha$, $\alpha_0$, and $\kappa$, we sweep over five values while keeping the other two fixed at their default:

$$\alpha \in \{0.1, 0.5, 2.0, 5.0, 10.0\},$$
$$\alpha_0 \in \{0.0, 0.1, 0.5, 1.0, 2.0\},$$
$$\kappa \in \{0, 5, 20, 50, 100\}.$$

The resulting RoboTwin-2 success rates are shown in Fig. 8. Across all three sweeps, performance peaks near the default values (red dots) and degrades when the prior is either too weak or too strong. Very small $\alpha$ or $\alpha_0$ encourage collapse onto a few experts, while very large values over-regularize the gates toward uniform or global usage. For $\kappa$, the worst case is $\kappa = 0$ (no stickiness), corresponding to the "W/o sticky gate" variant, whereas moderate stickiness ($\kappa \approx 20$–$50$) yields the best results; extremely large $\kappa$ makes gates overly inertial and slightly reduces performance.

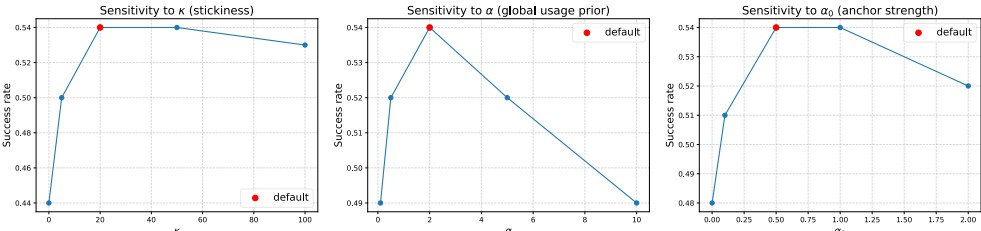

Figure 8: **Sensitivity of sticky-gate hyperparameters on RoboTwin-2 multi-task success.** Each curve varies a single parameter while keeping the others fixed at $(\alpha, \alpha_0, \kappa) = (2.0, 0.5, 20.0)$. Left: sweep over $\kappa$ (stickiness). Middle: sweep over $\alpha$ (global usage prior). Right: sweep over $\alpha_0$ (anchor strength). Red dots mark the default setting used in all main experiments.

Overall, these ablations show that the sticky Dirichlet–Markov gate is an essential but not overly fragile component of SMP: removing it significantly harms performance, while within the sticky family the method is robust around the chosen defaults and performs best when global usage, anchoring, and stickiness are all set to moderate levels.

## C.2 Ablation of Adaptive Expert Activation

SMP does not activate all experts at every timestep. Given the router mean $\bar{g}_t \in \Delta^{K-1}$, we define a per–expert mass $m_i = \bar{g}_{t,i}^2$ and greedily add experts in descending $m_i$ until the cumulative mass exceeds a coverage threshold $\tau_m$ or a hard top-$k$ budget is reached. This *adaptive expert activation* lets the policy use more experts on difficult states and fewer on simple ones; on RoboTwin-2, the default setting $\tau_m = 0.95$ activates on average $\approx 2.3$ experts per step.

We first remove this mechanism and always activate a fixed number of experts, setting top-$k = 4$ at all timesteps ("W/o adaptive expert" in Table 7). All other components (state-adaptive basis, sticky gates, diffusion experts) are unchanged. This variant performs slightly worse than full SMP on both multi-task and transfer benchmarks (e.g., RoboTwin-2 success drops from $0.54$ to $0.53$), showing that allowing the number of active experts to vary with the state gives a small but consistent benefit.

Next we keep the adaptive selection rule but change the mass definition from the quadratic form $m_i = \bar{g}_{t,i}^2$ to a linear one $m_i = \bar{g}_{t,i}$ ("Linear mass adapt." in Table 7). This also underperforms SMP (RoboTwin-2 success $0.52$), suggesting that the quadratic mass is helpful: it emphasizes high-confidence experts and suppresses mediocre ones, yielding sharper active sets, whereas the linear mass keeps medium gates relatively heavy and tends to recruit more partially relevant experts.

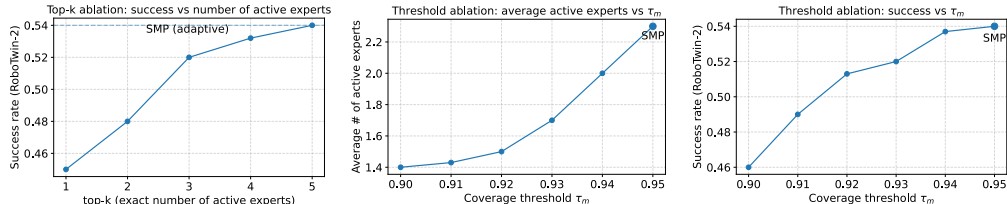

Figure 9: **Ablation of adaptive expert activation on RoboTwin-2.** Left: success rate when a fixed number of experts is used (hard top-$k$, no coverage rule). Middle: average number of active experts versus coverage threshold $\tau_m$ under adaptive activation; the SMP default ($\tau_m = 0.95$) uses about 2.3 experts on average. Right: success rate versus $\tau_m$, showing performance improving as more experts are recruited and saturating near the SMP configuration.

Finally, we perform a more fine-grained sensitivity study over (i) the hard top-$k$ and (ii) the coverage threshold $\tau_m$. In the first sweep we disable coverage and activate exactly $k \in \{1, 2, 3, 4, 5\}$ experts per step. The left panel of Fig. 9 shows that success on RoboTwin-2 increases monotonically from $0.45$ (top-1) to $0.54$ (top-5), with a large gap between $k = 1, 2$ and $k \geq 3$, and saturation once $k \geq 4$, which matches the full SMP model.

In the second sweep we restore adaptive activation and vary the coverage threshold while keeping all other settings fixed. As shown in the middle and right panels of Fig. 9, increasing $\tau_m$ from $0.90$ to $0.95$ increases the average number of active experts from about $1.4$ to $2.3$, while the success rate rises from roughly $0.46$ and smoothly saturates around $0.54$. When $\tau_m$ is too small, the model under-activates experts and behaves similarly to low top-$k$; beyond the default, adding more experts yields little gain. Overall, these ablations indicate that SMP benefits from recruiting a small but state-dependent set of experts, and that our default choice of $\tau_m$ lies in a regime where performance is high while the number of active experts remains modest.

## C.3 Skill Basis Selection

In SMP, actions are represented in a state-dependent orthonormal skill basis $B(s_t) \in \mathbb{R}^{d \times K}$, so that the decoded action is $a_t = B(s_t)\big(g_t \odot z_t\big)$. This basis changes smoothly with the robot state and task context, and its columns typically align with meaningful motion primitives such as translations and rotations of the end-effector. In multi-task settings, this state adaptivity is important: the "same" high-level skill (e.g., pushing, lifting, rotating) must manifest differently depending on arm pose, contact configuration, and the current object.

To assess the importance of this state dependence, we first replace the state-based basis with a *static* skill basis that does not depend on $s_t$. Concretely, we learn a single global orthonormal matrix $B$ and

use it for all states and tasks, while keeping the gates, diffusion experts, and activation mechanism unchanged. As shown in Table 7 (row "Fixed skill basis"), this severely degrades performance: on RoboTwin-2 the multi-task success rate drops from $0.54$ (SMP) to $0.40$, and the decline is even more pronounced in transfer settings (few-shot learning and skill composition). This suggests that a state-free basis cannot align skill directions with the local manipulation geometry and forces the MoE to work in a poorly matched coordinate system.

We then consider a PCA-based variant ("PCA skill basis" in Table 7), where we compute $K$ principal components of the action dataset and fix $B$ to this global PCA basis. This also performs poorly and, in our experiments, can come close to collapse: multi-task performance on RoboTwin-2 falls further to $0.32$, with similar drops on RLBench-2 and transfer tasks. Although PCA provides an orthogonal low-rank decomposition, it is global and task-agnostic; the resulting basis cannot adapt to specific tasks or phases, so different behaviors are entangled in the same components. Together, these ablations confirm that a state-dependent skill basis $B(s_t)$ is crucial for disentangling skills and achieving strong performance in multi-task and transfer robot manipulation.

## C.4 SCALING DP-STYLE BASELINES

A remaining question is whether the lower performance of DP-style baselines in multi-task settings is primarily due to under-parameterization, and whether scaling them to similar capacity would close the gap to SMP. To provide direct empirical evidence, we conduct controlled scaling experiments on DP, DP3, and ACT under the same RoboTwin-2 multi-task learning protocol, keeping the data, training schedule, and evaluation procedure fixed, and only increasing model capacity. In addition, we report inference time to quantify the accuracy–efficiency trade-off as capacity grows.

**Scaling DP/DP3/ACT to ∼300M parameters.** We scale each baseline to approximately 300M parameters (roughly $2\times$ the original size) and compare the average multi-task success rate (SR) and inference time on RoboTwin-2.

Table 8: Effect of scaling DP-style baselines to ∼300M parameters on RoboTwin-2 multi-task learning. We report success rate (SR) ↑ and inference time ↓.

| Method | Original SR | Scaled (∼300M) SR | Original Time | Scaled Time |
|--------|-------------|-------------------|---------------|-------------|
| DP | 0.29 | 0.37 | 120 ms | 160 ms |
| DP3 | 0.33 | 0.40 | 122 ms | 167 ms |
| ACT | 0.34 | 0.42 | 94.8 ms | 135 ms |

Scaling improves all three baselines, confirming that capacity matters. However, even at ∼300M parameters, these models remain below SMP ($0.54$ SR). Moreover, scaling increases inference time for all baselines. In contrast, SMP attains higher success with sparse activation (258M total parameters but only ∼ 80M activated at inference) and ∼ 107 ms inference time, yielding a more favorable accuracy–efficiency trade-off in multi-task multimodal manipulation.

