# OpenReview forum: "Abstracting Robot Manipulation Skills via Mixture-of-Experts Diffusion Policies"
_ICLR.cc/2026/Conference — ICLR 2026 Poster_

### Official Review · Reviewer_zQHP · 2025-10-22

**Soundness:** 3
**Presentation:** 2
**Contribution:** 3
**Rating:** 6
**Confidence:** 3

**Summary:**

The paper introduces SMP, a diffusion-based Mixture-of-Experts (MoE) policy designed to improve efficiency and generalization in multi-task manipulation. Unlike the existing MoE approaches, where the experts are entangled, the proposed method learns to route a subset of expert skills for each task, thereby reducing inference cost and enabling transferable skill learning across tasks. The authors provide theoretical analysis supporting the framework and conduct extensive experiments in both simulated and real-world environments to demonstrate its effectiveness. Overall, the paper contributes a novel integration of diffusion models and expert routing mechanisms for scalable and generalizable robotic control.

**Strengths:**

- The method is theoretically well-grounded, using a principled variational objective to learn a state-adaptive orthogonal skill basis.
- The paper's core innovation is combining this orthogonal basis with "sticky" routing dynamics. This is a novel approach that directly addresses prior limitations of skill entanglement and unstable, "chattering" gates.
- The approach shows significant empirical success, achieving higher multi-task and transfer learning performance than strong baselines.
- The method demonstrates markedly lower inference cost and latency by adaptively activating only a small subset of experts, highlighting its practical value for real-time control.

**Weaknesses:**

- The paper claims that a fixed basis "may fail to capture" action variability, which is the main justification for using a more complex state-adaptive basis. It would be better to include an ablation study comparing the proposed method to a version with a fixed basis. This would provide stronger evidence for this key design choice.
- The paper introduces some advanced mathematical concepts, like "thin-QR retraction with sign stabilization" and "sticky Dirichlet Markov dynamics". These might be a little unfamiliar to some readers. It would be helpful to add a brief intuitive explanation or a few more citations in the main text to help people understand the paper more smoothly.

**Questions:**

- For the inference time results in Table 2, your approach, SMP, uses the fewest active parameters during inference (80.2M), but its inference time (107.3 ms) is not the smallest; the ACT baseline is faster (94.8 ms). Do you have any insight on this? Is this slight overhead caused by other parts of your model, such as the router, the state-adaptive basis generation, or the final action composition step?
- The authors stated that the DP, DP3, and ACT baselines "underfit the multimodal distributions". In Table 2, the total number of parameters for these baselines (83.9M - 132.5M) is much smaller than the total parameters for your SMP method (258.9M). Is it possible that this "underfitting" is simply because the baseline models are under-parameterized in training space? What would you expect the success rate to be if these baselines were scaled up to have a similar total parameter count as SMP?

---

> ### Author Response · Authors · 2025-11-24
> **Response to Reviewer zQHP: Part 1/2**
>
> **Q1: Explain why SMP’s inference time is slightly slower than ACT despite using fewer active parameters.**
>
> **Response:** Thank you for the question. The reason is that the model size is not the only factor that determines inference time. ACT is an auto-regressive (action chunking transformer) model, while SMP is based on a diffusion process. Diffusion models iteratively denoise to generate actions, whereas ACT can produce the action sequence in a single forward pass. As a result, ACT is slightly faster in wall-clock inference time despite SMP using fewer active parameters.
>
> ---
> ---
>
> **Q2: Justify the claim that DP, DP3, and ACT “underfit the multimodal distributions” given their smaller parameter counts, and clarify whether their lower performance could be due to under-parameterization—also, discuss how their success rates might change if scaled to a similar total parameter count as SMP.**
>
> **Response:** This is a good point. To clarify, model capacity does affect performance to some extent, but our results indicate that capacity alone does not explain the gap between DP-style architectures and SMP.
>
> RDT provides a useful comparison point here. It is a monolithic diffusion policy with roughly 5x more parameters than SMP and roughly 10x more than DP and DP3. Yet its success rates do not surpass SMP (Table 1). This suggests that simply increasing parameter count does not resolve the core limitations of DP-style models in multi-task settings. The issue we refer to as "underfitting the multimodal distributions" stems from the structure of these policies rather than their size. DP, DP3, and ACT attempt to model all tasks through a single undifferentiated action distribution. In multi-task domains with diverse and multimodal behaviors, this architecture may fail to capture task or skill specific structure (the modes), even when scaled to larger parameter counts.
>
> SMP, in contrast, introduces an explicit skill mixture of experts. This provides an inductive bias for representing distinct modes and allows the policy to allocate capacity where it is most needed. As a result, SMP achieves high success rates with significantly fewer parameters than RDT. The few-shot transfer results provide further support for the role of structure rather than scale; Table 3 shows performance gap between RDT and SMP becomes even larger in this setting. We hope this clarification addresses the concern and we are happy to elaborate further if needed.

---

> > ### Author Response · Authors · 2025-11-24
> > **Response to Reviewer zQHP: Part 2/2**
> >
> > **Q3: Improve the accessibility of advanced mathematical components (e.g., thin-QR retraction with sign stabilization and sticky Dirichlet Markov dynamics) by adding intuitive explanations and/or more citations in the main text.**
> >
> > **Response:** Thank you for pointing this out. We agree that terms such as “thin-QR retraction with sign stabilization” and “sticky Dirichlet Markov dynamics” may be unfamiliar to many readers. In the revised manuscript, when these components are first introduced in the main text, we now explicitly direct readers to Appendix A.1 for a detailed and intuitive explanation.
> >
> > Appendix A.1 has been expanded to provide a more accessible treatment of both ingredients: (i) the geometric intuition behind the thin-QR retraction, including why sign stabilization is needed to avoid discontinuous flips in the skill basis, and (ii) an informal description of the sticky Dirichlet Markov dynamics and how they shape the temporal behavior and global usage of experts. This way, the main paper remains concise, while Appendix A.1 offers a self-contained, reader-friendly explanation for those who want to understand these mathematical components in more depth.
> >
> > ---
> > ---
> >
> >
> > **Q4: Provide empirical evidence for the benefit of a state-adaptive basis by adding an ablation comparing SMP to a variant with a fixed basis, directly testing the claim that a fixed basis may fail to capture action variability.**
> >
> > **Response:** We have added an explicit ablation in the revised manuscript that directly compares SMP with a learned **state-adaptive** basis $B(s_t)$ to a variant that uses a **fixed**, state-independent basis. In this ablated model, the basis is a single global orthonormal matrix $B$ shared across all states and tasks; all other components (sticky gates, diffusion experts, adaptive expert activation) are kept identical. This setup isolates the effect of state adaptivity in the basis.
> >
> > The results, summarized below and detailed in Appendix C.3 and Table below, show a clear degradation when the basis is fixed:
> >
> > | Method                      | RoboTwin-2 (Multi-task) | RLBench-2 (Multi-task) | Few-shot Transfer | Skill Composition |
> > |----------------------------|------------------------:|------------------------:|------------------:|------------------:|
> > | SMP (state-adaptive $B(s_t)$) | 0.54                     | 0.18                     | 0.38             | 0.30             |
> > | Fixed skill basis          | 0.40                     | 0.14                     | 0.31             | 0.24             |
> >
> > The fixed-basis variant consistently underperforms SMP across all benchmarks, with especially large gaps in the multi-task and transfer settings. This supports our claim that a fixed basis fails to capture the action variability induced by changing robot states and task phases, whereas the state-adaptive orthonormal basis $B(s_t)$ can re-orient skill directions with the manipulation geometry and thus yields significantly better performance. We now report and discuss this ablation explicitly in Appendix C.3 of the revised manuscript.

---

> > > ### Comment · Reviewer_zQHP · 2025-11-26
> > >
> > > Thank you for the detailed clarifications. Most of my concerns have been addressed. I still have one remaining question regarding Q2. I understand the argument that SMP should perform better in principle due to its structured skill basis. However, the empirical comparison between SMP and RDT does not necessarily show that SMP would also outperform DP, DP3, or ACT if those models were scaled to a similar capacity. Since RDT differs structurally from DP-style models, its weaker performance may not fully reflect what a larger DP/DP3/ACT variant would achieve.
> > >
> > > I agree with the authors’ intuition, but based on the current results, it is not clear whether this trend is empirically supported or remains a hypothesis. Any additional evidence or discussion on this point would be helpful.
> > >
> > > Overall, the revisions meaningfully improve the paper, and I am open to raising my score accordingly once this last point is clarified.

---

> > > > ### Author Response · Authors · 2025-12-01
> > > > **Reponse to Reviewer zQHP for Q2 and appreciation for rasing the score**
> > > >
> > > > Thank you for the thoughtful follow-up. We agree that our previous SMP vs. RDT comparison does not, by itself, settle what would happen if DP-style methods were scaled up, since RDT is structurally different. To directly address this, we conducted additional controlled scaling experiments on DP/DP3/ACT in the same RoboTwin-2 multi-task learning setting.
> > > >
> > > > ### Scaling DP / DP3 / ACT to ~300M parameters (and measuring inference time)
> > > > We scale each baseline to ~300M parameters (roughly **2×** their original size) and report both average multi-task success rate (SR) and inference time:
> > > >
> > > > | Method | Original SR | ~300M SR | Original inference time | ~300M inference time |
> > > > |---|---:|---:|---:|---:|
> > > > | DP  | 0.29 | 0.37 | 120 ms | 160 ms |
> > > > | DP3 | 0.33 | 0.40 | 122 ms | 167 ms |
> > > > | ACT | 0.34 | 0.42 | 94.8 ms | 135 ms |
> > > >
> > > > These results show that increasing capacity **does** improve DP-style diffusion policies and the transformer policy. **However**, even at ~300M parameters, the scaled variants still fall short of **SMP (0.54 SR)**. In addition, scaling increases inference time across all baselines, whereas **SMP achieves higher success with only ~80M activated parameters (out of 258M total) and ~107 ms inference time**, providing a stronger accuracy–efficiency trade-off than the enlarged dense baselines.
> > > >
> > > > **Summary.** In response to the concern that DP/DP3/ACT may simply be under-parameterized, we performed controlled scaling studies on RoboTwin-2. While increasing capacity consistently improves DP-style and transformer policies (e.g., scaling to ~300M yields higher SR across DP/DP3/ACT), the scaled variants still fall short of SMP’s success rate, and do so with higher inference time. Taken together, these results provide empirical evidence that the gap is not explained by parameter count alone, and that SMP’s structured, sparsely activated skill-MoE design delivers a more favorable accuracy–efficiency trade-off for multi-task multimodal manipulation.
> > > >
> > > > We have updated the manuscript in Appendix C.4 to include these scaling results and discussion.

---

### Official Review · Reviewer_kdN5 · 2025-10-27

**Soundness:** 3
**Presentation:** 2
**Contribution:** 3
**Rating:** 4
**Confidence:** 4

**Summary:**

This paper proposes Skill Mixture of Expert Policy (SMP), which learns a compact orthogonal skill basis, trains a mixture-of-experts diffusion policy on each basis, and employs a sticky routing mechanism. The main idea lies in decomposing the action space into orthogonal bases using differentiable QR factorizations to learn such bases, and establishing Dirichlet–Markov dynamics to control the gating. The results surpass the baselines and demonstrate a clear skill routing phase.

**Strengths:**

1. This paper proposes learning a lightweight basis in the action space and introduces a novel method for learning the gating mechanism. The experiments demonstrate the effectiveness of this approach in terms of success rate and inference computation (activated parameters). The paper also presents explainable and stable routing phase transitions, such as rotation and translation.
2. This paper exhibits several appealing features resulting from its decoupled structure (action basis, routing, and coefficients on the basis), including flexible inference sampling strategies and skill composition capabilities.

**Weaknesses:**

1. The paper claims that “a fixed basis may fail to capture such variability.” However, there is no comparison provided between a fixed basis and a learned basis. Moreover, from Figures 2 and 3, the action bases appear to be fixed and simple—such as left/right translation and rotation—suggesting that a fixed basis might be sufficient. How does the learned basis actually change with state?
2. In Equation (3), there are three hyperparameters. How do these parameters influence the final results?
3. Section 4.1 lacks sufficient explanation. For example, the equation is correct only under certain conditions, and the variable B does not appear in the right hand side. It would be better to clearly state the underlying assumptions and derivations.
4. Section 4.2 mentions top-k or coverage selection. What is the comparison between these two approaches? As the number of active experts increases, does the performance consistently improve?
5. The paper introduces several mathematical notations—such as simplex weights and Dirichlet–Markov dynamics—but does not explain them clearly. Including more mathematical details or visual illustrations would help improve clarity.
6. The paper would benefit from clearer writing and more detailed mathematical explanations.

**Questions:**

1. In Appendix B.3, the vision feature is concatenated with the robot state and projected into a shared feature o. Do the gating and basis modules also use this shared feature, or only the robot state? Additionally, could the authors provide details on the number of parameters and inference time for each component—namely, the vision encoder, gating module, basis, and diffusion experts?
2. Could the authors visualize the learned orthogonal basis throughout task execution? Figure 3 shows the basis directions, but they appear consistent within the same task execution, suggesting they may not vary with state.
3. It is quite common to apply PCA to extract principal components in dexterous manipulation and vehicle trajectory prediction, where PCA also learns an orthogonal basis. What advantages does learning a state-dependent basis provide compared to PCA or other fixed orthogonal decompositions?

---

> ### Author Response · Authors · 2025-11-24
> **Response to Reviewer kdN5: Part 1/3**
>
> **Q1: In Appendix B.3, the vision feature is concatenated with the robot state and projected into a shared feature o. Do the gating and basis modules also use this shared feature , or only the robot state?**
>
>
> **Response:** Thank you for pointing out this ambiguity. To clarify, the symbol $o$ in Appendix B.3 is a typo; it should refer to the shared observation feature $s_t$, not a separate variable.
>
> In the actual implementation, we follow the standard encoder used in diffusion-based visuomotor policies: a ResNet-18 processes the RGB observations together with the robot state and outputs a single fused feature vector $s_t$. This shared feature $s_t$ is then used as the input to all downstream components: the gating network, the state-adaptive skill-basis network, and the diffusion experts.
>
> We have corrected the notation and updated the description of the observation encoder in Appendix B.3 of the revised manuscript to clearly reflect this design.
>
> ---
> ---
>
> **Q2: Provide details on the number of parameters and inference time for each component—namely, the vision encoder, gating module, basis, and diffusion experts.**
>
> **Response:** Thank you for the suggestion. We have added a detailed breakdown of the number of parameters and per-step inference time for each component of SMP in the revised Appendix B.3. The statistics are summarized below:
>
> | Module                    | # Parameters | Inference time |
> |---------------------------|-------------:|---------------:|
> | Vision encoder            | 11.2M        | 23 ms          |
> | Gate                      | 3.6M         | 10 ms          |
> | Skill-basis network       | 12.5M        | 24 ms          |
> | Diffusion expert (single) | 28.9M        | 74 ms          |
>
> These values, together with the inference pipeline illustration, are now reported and discussed in Appendix B.3 of the revised manuscript.
>
> ---
> ---
>
> **Q3: Improve the mathematical clarity and exposition of Section 4.1, including explicit assumptions for the equations (and the role of B), and clearer explanations/illustrations of simplex weights and Dirichlet–Markov dynamics.**
>
> **Response:** Thank you for pointing this out. We have revised the paper to make the mathematical assumptions and notation in Section 4.1 clearer. Due to space constraints, we have placed these explanations in the Appendix.
>
> First, we added a new Appendix A.1 on *Mathematical preliminaries*, where we formally define simplex-valued gates $g_t \in \Delta^{K-1}$, the Dirichlet distribution, and the Dirichlet–Markov (“sticky”) dynamics used for the gate prior. We also provide an explicit description of the thin–QR retraction with sign stabilization used to construct the orthonormal skill basis $B(s_t)$, so that the role and properties of $B$ are clearly specified.
>
> Second, we added Appendix A.2 on *Subspace decomposition and variational inference*, which states the subspace-decomposition assumption under which the decoder equation is valid. In particular, we show that any action $a_t$ can be decomposed as
> $$
> a_t = B(s_t) u_t + R(s_t) r_t,
> $$
> for an orthonormal basis $B(s_t)$ and its complement $R(s_t)$, and that we identify $u_t = g_t \odot z_t$ while treating the residual term $R(s_t) r_t$ as small and absorbed into the Gaussian likelihood
> $$
> p(a_t \mid g_t, z_t, s_t, B) = \mathcal{N}\big(B(s_t)(g_t \odot z_t), \sigma_a^2 I\big).
> $$
> This makes explicit the conditions under which the decoder equation holds and clarifies why $B$ appears in the likelihood but not on the right-hand side of some simplified expressions in the main text.
>
> Finally, Appendix A.2 also contains the full derivation of the ELBO used in Section 4.1, based on this subspace decomposition and the sticky Dirichlet–Markov prior over gates. Together, these additions clarify the assumptions behind the equations in Section 4.1 and provide clearer explanations of $B$, simplex weights, and Dirichlet–Markov dynamics.

---

> ### Author Response · Authors · 2025-11-24
> **Response to Reviewer kdN5: Part 2/3**
>
> **Q4: Clarify and empirically justify the benefits of the learned state-dependent orthonormal basis over fixed or PCA-style bases, including: how the basis actually varies with state over a trajectory, visualizations of this variation, and comparisons against fixed/PC-based orthogonal decompositions.**
>
> **Response:** The revised manuscript now clarifies both how the state-dependent basis behaves and how it compares empirically to fixed and PCA-style bases.
>
> In SMP, actions are decoded as $a_t = B(s_t)\big(g_t \odot z_t\big)$, where $B(s_t)$ is an orthonormal basis that depends on the current robot state (e.g., robot end-effector configuration and object pose in the image). This lets the same high-level skill (such as “translation along a handle” or “rotation about a local axis”) re-orient with the scene geometry. In multi-task manipulation, the relevant motion directions depend strongly on $s_t$, so a state-dependent basis is important for capturing task- and phase-specific behavior.
>
> We have clarified the interpretation of Figure 3 in the paper. The figure visualizes the columns of $B(s_t)$ along a trajectory: translation- and rotation-like directions keep consistent semantics but rotate smoothly as the robot moves and the contact configuration changes. The basis is therefore not fixed in a global frame; it adapts with the state while staying orthonormal, which is precisely the intended “skill frame” behavior.
>
> To empirically justify this design, Appendix C.3 introduces an ablation where only the basis is changed and all other components of SMP (sticky gates, diffusion experts, adaptive expert activation) are kept identical. We consider:
>
> - **Fixed skill basis:** a single global orthonormal matrix $B$ shared by all states and tasks (no $s_t$ dependence).
> - **PCA skill basis:** a fixed orthonormal matrix $B_{\text{PCA}}$ formed by the top-$K$ principal components of the action dataset.
>
> The corresponding success rates are:
>
> | Method             | RoboTwin-2 (Multi-task) | RLBench-2 (Multi-task) | Few-shot Transfer | Skill Composition |
> |--------------------|-----------------:|-----------------:|------------------:|------------------:|
> | SMP (state-dependent $B(s_t)$) | 0.54 | 0.18 | 0.38 | 0.30 |
> | Fixed skill basis  | 0.40            | 0.14            | 0.31             | 0.24             |
> | PCA skill basis    | 0.32            | 0.11            | 0.26             | 0.20             |
>
> (See Appendix C.3 for details.)
>
> Both fixed and PCA-style bases cause substantial performance drops across multi-task learning and transfer. The PCA basis, although orthogonal and data-driven, is global and task-agnostic; it cannot adapt with $s_t$, so different tasks and phases are entangled in the same components and skills become hard to disentangle, sometimes approaching collapse. The fixed learned basis is slightly better but still clearly worse than the state-dependent version.
>
> Overall, the ablations and the trajectory visualizations in Figure 3 together show that a learned state-dependent orthonormal basis $B(s_t)$ is essential: it lets skill directions track the manipulation geometry over time and clearly outperforms fixed and PCA-based orthogonal decompositions in multi-task and transfer settings. Appendix C.3 reports and discusses these comparisons in the revised manuscript.

---

> ### Author Response · Authors · 2025-11-24
> **Response to Reviewer kdN5: Part 3/3**
>
> **Q5: Clarify how the sticky gating function’s (Eq. 3) hyperparameters affect performance, and compare top-k versus coverage-based expert selection, including how performance scales with the number of active experts.**
>
>
> **Response:** We have added explicit sensitivity experiments for both the sticky gating hyperparameters and the expert-activation mechanism in Appendices C.1 and C.2. Here we summarize the quantitative trends on RoboTwin-2.
>
>
> ### Effect of sticky gating hyperparameters
>
> The sticky gate prior in Eq. (3) is parameterized by $(\alpha, \alpha_0, \kappa)$, controlling the global usage prior, the anchor strength to the global usage, and the temporal stickiness, respectively. In all main experiments, we use the default $(\alpha, \alpha_0, \kappa) = (2.0, 0.5, 20)$, which gives a success rate of about $0.54$ on RoboTwin-2. In Appendix C.1 we vary each parameter independently while fixing the others at their defaults. The measured success rates are:
>
> - Varying $\alpha$ (global usage prior):
>
>   | $\alpha$      | 0.1  | 0.5  | **2.0** | 5.0  | 10.0 |
>   |---------------|:----:|:----:|:-------:|:----:|:----:|
>   | Success       | 0.49 | 0.52 | **0.54**| 0.52 | 0.49 |
>
> - Varying $\alpha_0$ (anchor strength):
>
>   | $\alpha_0$    | 0.0  | 0.1  | **0.5** | 1.0  | 2.0  |
>   |---------------|:----:|:----:|:-------:|:----:|:----:|
>   | Success       | 0.48 | 0.51 | **0.54**| 0.54 | 0.52 |
>
> - Varying $\kappa$ (stickiness):
>
>   | $\kappa$      | 0    | 5    | **20**  | 50   | 100  |
>   |---------------|:----:|:----:|:-------:|:----:|:----:|
>   | Success       | 0.44 | 0.50 | **0.54**| 0.54 | 0.53 |
>
> These results show that performance peaks near the default setting and degrades smoothly at extreme values. In particular, $\kappa = 0$ (no temporal coupling) corresponds to the non-sticky case and yields the lowest success (0.44). Moderate ranges around $(\alpha, \alpha_0, \kappa) = (2.0, 0.5, 20)$ all maintain high performance, and we do not observe collapse in any of these settings.
>
>
> ### Top-$k$ versus coverage-based expert selection and scaling with active experts
>
> Appendix C.2 analyzes how performance scales with the number of active experts and how coverage-based selection compares to hard top-$k$.
>
> 1. **Hard top-$k$ (no coverage):**
>    If we disable the coverage rule and always activate exactly $k$ experts at each timestep, the RoboTwin-2 success rate behaves as:
>
>    | top-$k$        | 1    | 2    | 3    | 4      | 5      |
>    |----------------|:----:|:----:|:----:|:------:|:------:|
>    | Success        | 0.45 | 0.48 | 0.52 | 0.53  | 0.54   |
>
>    Performance increases as $k$ grows and essentially saturates once $k \ge 4$, matching the full SMP success of 0.54. The “W/o adaptive expert” configuration in Table 7 corresponds to the special case top-$k = 4$, which indeed sits slightly below SMP (0.53 vs 0.54).
>
> 2. **Coverage-based activation: threshold, number of experts, and success:**
>    With adaptive activation enabled, we use a coverage threshold $\tau_m$ and a quadratic mass $m_i = \bar g_{t,i}^2$. Sweeping $\tau_m \in \{0.90, 0.91, 0.92, 0.93, 0.94, 0.95\}$ yields:
>
>    | $\tau_m$              | 0.90 | 0.91 | 0.92  | 0.93 | 0.94 | 0.95 |
>    |-----------------------|:----:|:----:|:-----:|:----:|:----:|:----:|
>    | Avg. active experts   | 1.40 | 1.43 | 1.50  | 1.70 | 2.00 | 2.30 |
>    | Success               | 0.46 | 0.49 | 0.513 | 0.52 | 0.537| 0.54 |
>
>    As $\tau_m$ increases, the average number of active experts grows from about 1.4 to 2.3, and the success rate increases correspondingly from 0.46 to 0.54, then saturates. Low thresholds under-activate experts and behave similarly to low top-$k$; the default $\tau_m = 0.95$ lies near a plateau where performance is high while the active set remains small (about 2–3 experts).
>
> 3. **Quadratic versus linear mass:**
>    At the same threshold, using linear mass $m_i = \bar g_{t,i}$ instead of quadratic mass $m_i = \bar g_{t,i}^2$ (“Linear mass adapt.” in Table 7) reduces RoboTwin-2 success from 0.54 to 0.52. This suggests that the quadratic mass improves expert selection by emphasizing high-confidence experts and suppressing weaker ones, leading to more effective sparse activation.
>
>
>
> In summary, the sticky gate hyperparameters $(\alpha, \alpha_0, \kappa)$ have a clear but smooth effect on performance, with a broad region of good settings around our default choice. On the activation side, both the number of active experts and the selection rule matter: success improves as the model can recruit about 2–3 experts per state and then saturates, and coverage-based selection with quadratic mass provides a better trade-off between performance and sparsity than either fixed top-$k$ or linear mass.

---

> > ### Comment · Reviewer_kdN5 · 2025-11-27
> >
> > Thank you for your response and the revision. Since the paper is now clearer and more complete, I would like to raise my score.

---

> > > ### Author Response · Authors · 2025-11-27
> > > **Appreciation for Reviewer kdN5’s Feedback and Revised Score**
> > >
> > > We sincerely thank Reviewer kdN5 for the careful reading of our paper and the constructive feedback. We are very glad that the revisions and additional experiments have helped clarify the presentation and strengthen the empirical support for our approach.
> > >
> > > Your comments on the mathematical exposition, the role of the state-dependent basis, and the behavior of the sticky gating mechanism were especially helpful in improving the clarity and completeness of the manuscript. We appreciate your willingness to re-evaluate the submission in light of the revisions and to raise your score.
> > >
> > > We will further refine the writing and explanations in the camera-ready version to ensure that these clarifications are as accessible as possible to future readers. Thank you again for your time and thoughtful review.

---

### Official Review · Reviewer_aj4r · 2025-11-01

**Soundness:** 4
**Presentation:** 3
**Contribution:** 4
**Rating:** 8
**Confidence:** 4

**Summary:**

This paper introduces a diffusion-based MoE framework designed for reusable motion primitives across multiple tasks. It features in orthonormal skill basis and sticky routing that reduce gate switches and activated experts. Abundant experiments have been conducted both in simulation and real world with ablation studies supporting the design choices. Overall I think this is a solid submission for ICLR.

**Strengths:**

1. The paper is well-organized and easy to follow.
2. The idea of orthogonal skill basis seems very attractive, based on the ablation studies it does provide good cross-task skill reuse with few switches.
3. The ablation studies are abundant and helpful in supporting the claimed contributions.
4. Real-world experiments looks good.

**Weaknesses:**

1. I am curious about the training cost of learning good orthonormal skill basis, more details on this part will be appreciated.
2. In Figure 3 task "Put bread into skillet" and "Lift tray with block in it", there are still considerable portion of the trajectories with multiple experts activated at the same time with similar gate values, does it mean in these cases the experts are still overlapping?

**Questions:**

See weakness

---

> ### Author Response · Authors · 2025-11-24
> **Response to Reviewer aj4r**
>
> **Q1: Explain the training cost and efficiency of learning the orthonormal skill basis.**
>
> **Response:** Thank you for raising this point. We agree that the training cost of learning a good orthonormal skill basis is important for practical deployment, and we have added more details in the revised Appendix B.3. All SMP experiments are trained end-to-end on a single NVIDIA A6000 GPU. The table below reports the parameter count of each module:
>
> | Module                    | # Parameters |
> |---------------------------|-------------:|
> | Vision encoder            | 11.2M        |
> | Gate                      | 3.6M         |
> | Skill-basis network       | 12.5M        |
> | Diffusion expert (single) | 28.9M        |
>
> The skill-basis network therefore adds only a moderate number of parameters (12.5M) relative to the diffusion expert, so the additional training cost for learning the orthonormal basis is limited. For the multi-task experiments on RoboTwin 2.0 and RLBench 2, SMP is trained for 3000 epochs, corresponding to roughly 20–25 hours of wall-clock time on this GPU. Under this budget, the training losses converge reliably and the learned skill basis performs well across tasks. Appendix B.3 of the revised manuscript now reports this breakdown and discusses the associated computation cost.
>
> ---
> ---
>
> **Q2: In Figure 3 task "Put bread into skillet" and "Lift tray with block in it", there are still considerable portion of the trajectories with multiple experts activated at the same time with similar gate values, does it mean in these cases the experts are still overlapping?**
>
> **Response:** Yes, this observation is correct. In the “Put bread into skillet” task, both left and right arms translate and rotate simultaneously, so multiple experts are activated at the same time. This is also a consequence of our adaptive expert activation, which dynamically controls expert usage; if we only allowed a fixed top-k experts to be active, such complex behaviors might be under-represented.

---

### Official Review · Reviewer_fTZq · 2025-11-02

**Soundness:** 4
**Presentation:** 4
**Contribution:** 4
**Rating:** 8
**Confidence:** 4

**Summary:**

The paper proposes a mixture of experts framework for diffusion-based policy learning in multi-task training scenario. The main difference from a "standard MoE model for score function" approach is that the authors introduce a state-dependent orthonormal action basis, where each vector in the basis represents a skill. The weights per skill are predicted by skill-specific diffusion experts. A gating function is used to activate a set of experts and ensure that the gating weights do not change significantly for consecutive inferences. Strong results are shown on multi-task bimanual manipulation scenarios.

**Strengths:**

The paper presents an intuitive method for learning skills implicitly for multi-task scenarios. The engineering in choosing the suitable method to construct the orthonormal basis, training targets for skill-specific diffusion experts, and sticky gating function is well executed.

**Weaknesses:**

Overall the major concern I have is that the paper do not provide any ablation results for the specific design choices like:

1. How does the results change without the sticky gating function? Since this is one of the major contributions, it will be worth looking at how this imapcts simpler MoEs like Sparse DP.
2. How does the results change with k or the mass threshold? Linear mass vs quadratice mass?
3. The practical implementation subsection in the appendix suggests that the proposed method requires significant tuning to avoid collapse.

**Questions:**

See weaknesses above.

---

> ### Author Response · Authors · 2025-11-24
> **Response to Reviewer fTZq: Part 1/3**
>
> **Q1: How does the results change without the sticky gating function? Since this is one of the major contributions, it will be worth looking at how this imapcts simpler MoEs like Sparse DP.**
>
> **Response:** Thank you for pointing this out. We have added a dedicated ablation of the sticky gating mechanism in Appendix C.1. In SMP, the gate sequence $g_t \in \Delta^{K-1}$ is regularized by a sticky Dirichlet–Markov prior with a global usage vector $\vartheta$: $\vartheta \sim \mathrm{Dir}(\alpha \mathbf{1})$, $g_1 \sim \mathrm{Dir}(\alpha_0 \vartheta)$, $g_t \sim \mathrm{Dir}(\kappa g_{t-1} + \alpha_0 \vartheta)$ for $t \ge 2$. This prior encourages both balanced global usage (via $\vartheta$) and temporally smooth “sticky’’ expert assignments (via $\kappa g_{t-1}$).
>
> To measure its impact, we train a variant where this prior is removed and the router is a plain feedforward network that outputs gates independently at each step (“W/o sticky gate” in Table 7). All other components (state-adaptive basis, diffusion experts, adaptive expert activation) are unchanged. The resulting success rates are:
>
> | Method             | RoboTwin-2 (Multi-task) | RLBench-2 (Multi-task) | Few-shot Transfer | Skill Composition |
> |--------------------|:-----------------------:|:----------------------:|:-----------------:|:-----------------:|
> | **SMP (full)**     | **0.54**                | **0.18**               | **0.38**          | **0.30**          |
> | W/o sticky gate    | 0.44                    | 0.15                   | 0.33              | 0.26              |
>
> On RoboTwin-2, removing sticky gating reduces the success rate from 0.54 to 0.44 (about a 19% relative drop), and we observe similar degradations on RLBench-2 and both transfer benchmarks. Qualitatively, the non-sticky router exhibits high-frequency switching and tends to collapse onto a small subset of experts, behaviour similar to standard sparse MoE policies such as Sparse DP. In contrast, the sticky Dirichlet–Markov dynamics produce smooth, phase-like gate trajectories and more balanced expert usage, which leads to higher and more stable performance. These quantitative and qualitative comparisons are now documented in Appendix C.1.
>
> Regarding simpler FFN-based MoE architectures such as Sparse DP, we would like to clarify that the proposed sticky gating mechanism is **not** directly suitable in that setting. In SMP, the MoE is at the **policy level**: each expert is explicitly intended to represent a temporally extended skill, and the gate $g_t$ is a trajectory-level latent that should remain consistent over consecutive timesteps within a phase. In contrast, FFN-style MoEs use gates that are evaluated independently at each layer and (token/time) step, with no temporal structure and no requirement that “expert $i$ at time $t$” corresponds to the same computation or behavior at time $t{+}1$ (please see Fig 2 in the [Sparse DP paper](https://arxiv.org/pdf/2407.01531)). Their experts act as generic feature transformers rather than persistent skills. Imposing a sticky Dirichlet–Markov prior on such per-layer gates would therefore be misaligned with the architecture rather than a fair drop-in modification. For this reason, we focus our sticky gating analysis on the policy-level skill MoE in SMP, where temporally coherent experts are well-motivated and, as our ablations show, lead to clear performance gains.

---

> > ### Author Response · Authors · 2025-11-24
> > **Response to Reviewer fTZq: Part 2/3**
> >
> > **Q2: How does the results change with k or the mass threshold? Linear mass vs quadratice mass?**
> >
> > **Response:** We have added an ablation of the active-expert selection mechanism in Appendix C.2. By default, SMP uses an adaptive activation rule with quadratic mass $m_i = \bar g_{t,i}^2$ and a coverage threshold $\tau_m = 0.95$, which yields about 2.3 active experts per timestep on RoboTwin-2. The main variants are summarized below:
> >
> > | Method                          | RoboTwin-2 (Multi-task) | RLBench-2 (Multi-task) | Few-shot Transfer | Skill Composition |
> > |---------------------------------|:-----------------------:|:----------------------:|:-----------------:|:-----------------:|
> > | **SMP (adaptive, quad. mass)** | **0.54**                | **0.18**               | **0.38**          | **0.30**          |
> > | W/o adaptive expert¹           | 0.53                    | 0.18                   | 0.37              | 0.29              |
> > | Linear mass adapt.²            | 0.52                    | 0.17                   | 0.37              | 0.29              |
> >
> > ¹ fixed top-$k = 4$ at all timesteps.
> > ² same adaptive selection rule, but with $m_i = \bar g_{t,i}$ instead of $m_i = \bar g_{t,i}^2$.
> >
> > In the “W/o adaptive expert” variant, we disable the coverage rule and always
> > activate a fixed top-$k = 4$ experts at each timestep, while keeping all other
> > components identical. This yields a small but consistent drop compared to full
> > SMP (e.g., RoboTwin-2 from 0.54 to 0.53; see Table above). Appendix C.2 further analyzes this via a hard top-$k$ sweep: when we disable coverage and set $k \in \{1,2,3,4,5\}$, the RoboTwin-2 success rate behaves as summarized below:
> >
> >
> > | Setting        | top-1 | top-2 | top-3 | top-4 | top-5 |
> > |----------------|:-----:|:-----:|:-----:|:-----:|:-----:|
> > | Success rate   | 0.45  | 0.48  | 0.52  | 0.53  | 0.54  |
> >
> >
> > The “W/o adaptive expert” configuration corresponds exactly to the top-4 case,
> > which lies slightly below the fully adaptive scheme. These results show that
> > while a reasonable fixed top-$k$ (e.g., $k \ge 3$) can approach the
> > performance of full SMP, adaptive expert activation still provides a modest but
> > reliable advantage.
> >
> >
> > In the “Linear mass adapt.” variant, we keep adaptive activation but change the mass from quadratic to linear, $m_i = \bar g_{t,i}$. This further reduces performance (e.g., RoboTwin-2 to 0.52). Intuitively, the quadratic mass $m_i = \bar g_{t,i}^2$ emphasizes high-confidence experts, producing sharper and more specialized active sets. The linear mass keeps medium gate values relatively "heavy", so the coverage rule tends to include more partially relevant experts, which slightly weakens specialization. These results support our design choice of using adaptive activation with a quadratic mass function in the main SMP model.

---

> > > ### Author Response · Authors · 2025-11-24
> > > **Response to Reviewer fTZq: Part 3/3**
> > >
> > > **Q3: The practical implementation subsection in the appendix suggests that the proposed method requires significant tuning to avoid collapse.**
> > >
> > > **Response:** We agree that, in practice, all methods (including the baselines) require some degree of hyperparameter tuning to reach their best performance, and SMP is no exception. However, our additional experiments show that SMP is **robust** to reasonable changes in its key hyperparameters and does **not** exhibit collapse or unstable behaviour under such variations.
> > >
> > > To make this clear, we have added systematic sensitivity studies in Appendix C.1 (for the sticky gating prior) and Appendix C.2 (for adaptive expert activation). In Appendix C.1, we vary the sticky-gate hyperparameters $(\alpha, \alpha_0, \kappa)$ over a wide range. Performance changes smoothly and only degrades significantly when the prior is essentially removed (e.g., very small $\kappa$, corresponding to the “W/o sticky gate’’ case), at which point the model behaves more like a standard sparse MoE. In all reasonable settings around our default $(2.0, 0.5, 20.0)$, the gates remain well-behaved and we do not observe mode collapse.
> > >
> > > In Appendix C.2, we study the expert-activation mechanism by (i) removing adaptive activation and using a fixed top-$k$, (ii) switching from quadratic mass $m_i = \bar g_{t,i}^2$ to linear mass $m_i = \bar g_{t,i}$, and (iii) sweeping both $k$ and the coverage threshold $\tau_m$. Across these ablations, success rates vary gradually and only drop sharply in clearly suboptimal regimes (e.g., top-1 or very low $\tau_m$ that severely under-activate experts). The default configuration with quadratic mass and $\tau_m = 0.95$ lies in a broad plateau where performance is high and the average number of active experts remains small.
> > >
> > > Taken together, these results show that while SMP, like other methods, benefits from tuning for optimal performance, it is not fragile: it remains stable and does not collapse under moderate hyperparameter changes. We have updated the implementation subsection to reflect this and now explicitly refer to the sensitivity analyses in Appendices C.1 and C.2.

---

### Author Response · Authors · 2025-12-02
**Genearal response to the Area Chair**

Dear Area Chair,

We would like to provide a brief **general response** to clarify the overall rebuttal process and how we addressed reviewer feedback across rounds. For transparency, we summarize updates in three parts: **(1) the initial review results**, **(2) Round 1 rebuttal** and manuscript revisions (before the author-info leak on Nov. 27), and **(3) Round 2 rebuttal** with additional experiments (after Nov. 27).

---

## 1) Initial review results (Round 0)
Our submission received four reviews:

- **Reviewer zQHP:** Rating **6**, Confidence **3**
- **Reviewer kdN5:** Rating  **4**, Confidence **4**
- **Reviewer aj4r:** Rating    **8**, Confidence **4**
- **Reviewer fTZq:** Rating  **8**, Confidence **4**

**Average rating:** **6.5**
**Average confidence:** **3.75**

Across the reviews, the main questions fell into four themes:
1. **Mathematical clarity** (introduction and derivations of key equations; more intuition for advanced components).
2. **Experimental / implementation clarification** (training cost, inference breakdown, protocol details, and reporting clarity).
3. **Ablation studies and sensitivity analyses** (sticky gating, activation rule/top-k/coverage threshold, linear vs. quadratic mass, stability).
4. **Evidence for state-adaptive vs. state-free (fixed/PCA) skill basis** (direct comparisons and how the basis varies with state).

---

## 2) Round 1 rebuttal (posted Nov. 24; before Nov. 27)
In Round 1, we updated the rebuttal on **Nov. 24** to address the four themes above and revised the manuscript accordingly:

1) **Improved mathematical clarity and accessibility.**
We expanded the appendix with clearer definitions, assumptions, and derivation details for the key equations, and added intuitive explanations for components such as the thin-QR retraction with sign stabilization and the sticky Dirichlet–Markov gating dynamics.

2) **Clarified experimental and implementation details.**
We added training-cost reporting and detailed component-level breakdowns (e.g., parameter counts and per-step inference-time contributions), and clarified the observation feature usage across modules (gating, basis, experts).

3) **Added ablation studies and hyperparameter sensitivity.**
We added systematic ablations for (i) removing sticky gating, (ii) removing adaptive expert activation (fixed top-k), and (iii) switching quadratic to linear mass, along with sensitivity sweeps for key hyperparameters.

4) **Added direct fixed/PCA basis comparisons.**
We added ablations comparing the learned state-adaptive basis against fixed (state-independent) and PCA-based orthogonal bases, holding other components unchanged.

After Round 1, Reviewer **kdN5** indicated that the revisions addressed their concerns and **raised their score to 6**. Reviewer **zQHP** also noted that most concerns were addressed and stated **he/she was open to raising their score** once we provided additional evidence regarding the effect of scaling DP/DP3/ACT model sizes. These replies were posted **before Nov. 27**, i.e., prior to the author-info leak discussion.

---

## 3) Round 2 rebuttal (after Nov. 27): new scaling experiments for DP/DP3/ACT
Following Reviewer zQHP’s remaining concern about whether DP/DP3/ACT were simply under-parameterized, we conducted **additional controlled scaling experiments** on RoboTwin-2 multi-task learning.

We scaled DP/DP3/ACT to ~ 300M parameters (≈2× their original sizes) and measured both success rate and inference time. Scaling improves baseline success rates, but the scaled models still underperform our SMP and incur higher inference time. In contrast, SMP achieves higher success while activating fewer parameters at inference (~ 80M activated out of 258M total) and lower latency (~107 ms) than the scaled dense baselines.

We updated the manuscript in Appendix C.4 to include these scaling results and corresponding discussion, **directly addressing the reviewer zQHP’s request** for empirical evidence rather than a hypothesis about scaling trends.

---

**Summary.** We believe we have **addressed the key concerns** raised in the initial reviews. In Round 1, we revised the manuscript to improve mathematical clarity, strengthen/clarify experimental reporting, and add the requested ablations (including fixed/PCA basis comparisons and sensitivity analyses). In Round 2, we conducted additional controlled scaling experiments for DP/DP3/ACT to directly test the under-parameterization concern, and incorporated these results into the appendix. Collectively, these updates provide stronger empirical and explanatory support for SMP and clarify its accuracy–efficiency advantages in multi-task multimodal manipulation.



Sincerely,
The Authors

---

### Meta-Review · Area_Chair_Qc5A · 2026-01-06

**Summary:**

The reviewers’ major concerns centered on (1) an insufficient explanation of the underlying mathematical principles of the method, and (2) a lack of clear motivation for why—and under what conditions—baseline approaches such as PCA-based or fixed basis may fail. In addition, some reviewers requested more experimental studies to demonstrate effectiveness, especially on cross-task skill transfer. The authors have addressed these concerns well, and I recommend acceptance.

**Reviewer Concerns:**

The reviewer kdN5 and zQHP's concerns about the mathematical principle explanation and the motivation studies are well addressed. The reviewer fTZq and kdN5's concerns about hyperparameter studies and ablation studies are addressed too. Besides, the reviewer zQHP's scaling&underfitting study suggestions and kdN5's visualization comments are interesting, the authors also addressed well. The reviewer aj4r pointed out that their cross-task skill experiments are good.

**Reviewer Scores:**

Reviewer kdN5 and zQHP fully participate the discussion and provide many good comments, the authors addressed their concerns well.

---

### Decision · Program_Chairs · 2026-01-26

Accept (Poster)